# Decision on optimal airflow regulation solution set based on heat and airflow coupling characteristics of mine airflow in time series

Tong Jia[1,2], Heng Ma [1,2]*, Ke Gao[1,2]

1 College of Safety Science and Engineering, Liaoning Technical University, Huludao, Liaoning, PR China, 2 Key Laboratory of Mine Thermodynamic Disasters and Control of Ministry of Education, Liaoning Technical University, Huludao, Liaoning, PR China

* mahenglgd@163.com

## Abstract

Addressing the issue where traditional airflow regulation models, which treat natural ventilation pressure and ventilation resistance as constant values, fail to determine the optimal solution set for airflow regulation based on real-time dynamic mine environments, this research proposes a model based on the characteristics of airflow's aerodynamic coupling with heat flow in an actual mine. A density-pressure-temperature characteristic equation is introduced to describe the variation laws of airflow state properties, forming a transient heat and airflow flow characteristics model in time series. Using the transient heat and airflow flow characteristics model as a correction variable, the fluctuating natural ventilation pressure and ventilation resistance model in time series is established. The fluctuating natural ventilation pressure and ventilation resistance are treated as environmental variables, and adjustment resistance is treated as a decision variable. A nonlinear correction airflow regulation model based on the heat and airflow coupling characteristics of airflow is constructed. This model is solved to obtain the optimal solution set for real-time airflow regulation, enabling precise airflow control. To verify the reliability of model, taking the Gucheng Coal Mine as an engineering research subject, An unsteady-state environmental field, which transfers energy to the airflow in a gradient flow field, is taken as the boundary condition. The unsteady-state heat transfer numerical model for the Gucheng Coal Mine is solved to analyze the heat and airflow coupling characteristics of airflow and to validate the proposed model. By inputting the heat environment time-series data of Gucheng Coal Mine into the transient heat and airflow flow characteristics model, the fluctuating natural ventilation pressure and ventilation resistance model, and the airflow regulation model, an optimal airflow control scheme is decided that meets the airflow requirements underground while minimizing system power consumption. The research provides a new theoretical foundation and methodological support for intelligent ventilation volume control in mines.

**Data availability statement:** All relevant data are within the paper.

**Funding:** The author(s) received no specific funding for this work.

**Competing interests:** "The authors declare that they have no competing interests."

## 1. Introduction

The advancement of the smart mining process has brought about improvements in the mechanized production of coal mines, and high-speed production will inevitably increase the scale and depth of mining [1–3]. This expansion not only requires an improvement in fan efficiency to cope with the enlarged production control area but also leads to dynamic changes in the ground stress of the roadway and the stability of the surrounding rock under the effects of the rock mass self-weight stress and stope stress disturbances, thereby altering the characteristics of the roadway resistance [4,5]. The changes in fan efficiency and the resistance characteristics of the roadway result in non-constant characteristics of the fan working conditions. The varying roadway resistance and the fan static pressure, as energy terms in the Bernoulli's equation, will cause a redistribution of the ventilation pressure in roadways, altering the distribution weights of the airflow in the ventilation network [6]. This leads to dynamic fluctuations in ventilation parameters (such as airflow, air pressure, and air velocity) over time.

Simultaneously, the expansion of production scale manifests as an increase in mining depth in the vertical space. Under the combined effects of geothermal flux, geological gradient, and airflow compression effect, a high-depth mine will form an unsteady heat environment field and a gradient flow field [7,8]. Within this gradient flow field, the mine airflow will undergo energy exchange primarily through heat exchange with the heat environment field, leading to the conversion of the airflow's internal energy and kinetic energy [9]. The airflow exhibits a non-isoheat flow state, with its thermodynamic and dynamic parameters undergoing interlinked changes. In other words, over temporal and spatial scales, the ventilation parameters and state properties (temperature, pressure, density, etc.) of the mine airflow change in a coupled manner, indicating that the airflow possesses heat and airflow coupling characteristics.

When the fluctuations in the ventilation parameters and state properties of the airflow exceed a certain range over time, from the perspective of working conditions, the match degree between the fan supplying air with constant blade angle and frequency parameters and the mine roadways' resistance calculated with constant frictional resistance decreases. This mismatch can easily lead to an imbalance in the supply and demand of air volume in the mine roadways, which can, in confined spaces, easily trigger mine disasters, threatening the safety of mine production [10,11]. Therefore, to avoid anomalies in the conservation of supply and demand air volume in the mine roadways, it is necessary to construct an airflow regulation model to quantitatively decide on a ventilation safety management plan, thereby achieving precise control of the air volume in the ventilation network.

Extensive research has been conducted on the development of technical methods for airflow control in coal mines, particularly along three major directions [12]: constructing mathematical models for airflow control, developing control algorithms for determining the adjustment parameter, and performing intelligent control of ventilation facilities. Yu et al [13]. established an airflow regulation model aimed at minimizing system power consumption and maximizing fan efficiency to improve the model's accuracy in describing complex ventilation networks. They enhanced the model's solution accuracy by improving the sorting rules of Pareto solutions. Nyaaba W et al [14]. Standardized the inequality constraints of the nonlinear programming model of the ventilation network by introducing slack variables and proposed the first-order Lagrangian algorithm to solve generalized nonlinear constraint problems. LI et al [15]. Proposed an improved bare-bones particle swarm optimization algorithm for solving the airflow regulation model, which optimizes the contraction factor, initial values, and cross-over particles to enhance population diversity and address the issue of premature convergence. These studies focus on the dynamic airflow process in mines, treating the airflow as an isochoric process characterized by constant density. They construct and solve airflow regulation models by considering natural ventilation pressure and ventilation resistance as constant environmental parameters, and regulate airflow based on its dynamic properties.

In 1986, Guo [16] proposed that the heat transfer process and the flow process of fluids are mutually controlling, and that the density changes caused by temperature gradients can affect the fluid flow resistance. He introduced the concept of "heat resistance". Guo [17] also pointed out that under significant heat exchange conditions, temperature gradients, pressure gradients, and density gradients can drive fluid flow, forming a "heat compressibility effect" that alters the pressure distribution in the flow field. In 1997, Yang [18] introduced the concept of "heat compressibility effect" into the field of mine ventilation, suggesting that mine airflow under the influence of multiple heat sources is not a one-dimensional steady flow but an unsteady flow under the coupled effects of heat and dynamic forces. Based on this, scholars have conducted research on the effects of heat flow on natural ventilation pressure and ventilation resistance in mine airflow, aiming to improve the calculation accuracy of natural ventilation pressure and ventilation resistance for airflow regulation. Most of these studies have focused on the heat transfer and flow of fire-induced airflow, with less attention to the heat flow under normal production conditions. In recent years, MA et al [19] and Zhang et al [20]. Studied the changes in internal energy and pressure loss caused by heat exchange between the airflow and the surrounding rock. They proposed the theory that airflow in an underground heat environment generates heat ventilation pressure and flow heat resistance. Nie et al [21]. Introduced the concept of heat ventilation pressure and extended the heat effects between airflow and the heat environment to the field of local natural ventilation pressure. These studies focus on the quantitative analysis of the impact of heat flow at a specific moment on natural ventilation pressure and ventilation resistance, but have not yet quantified and incorporated the effects of heat flow over time on natural ventilation pressure and ventilation resistance into airflow regulation models.

The natural ventilation pressure and ventilation resistance calculated based on the constant state properties of an isochoric process or the constant state properties at a specific moment are constant values in the time series. An airflow regulation model constructed with constant natural ventilation pressure and ventilation resistance as environmental parameters cannot determine the optimal solution set for airflow regulation to achieve precise control of the ventilation network air volume. Therefore, the key to quantitatively precise airflow control lies in solving the fluctuation values of airflow state properties over time based on the heat and airflow coupling characteristics of the airflow, calculating the fluctuating natural ventilation pressure and ventilation resistance, and accordingly, non-linearly correcting the airflow regulation model to solve for the optimal solution set. Based on this, the primary implementation methods for the research are proposed as follows: By analyzing the dynamic variation patterns of the state properties of mine airflow over time, construct a transient heat and airflow flow characteristics model and a fluctuating natural ventilation pressure and ventilation resistance model to solve for the fluctuating values of natural ventilation pressure and ventilation resistance over time. Use these fluctuating natural ventilation pressure and ventilation resistance as environmental variables, the adjustment resistance as decision variables, and the system power consumption minimum as the objective function to construct an airflow regulation model based on the heat and airflow coupling characteristics of the airflow. Through model calculation, obtain the optimal solution set for airflow regulation, ultimately achieving precise control of the ventilation network air volume, providing theoretical basis and technical support for enhancing the reliability of the mine ventilation system.

## 2. Airflow regulation mathematical model based on heat and airflow coupling characteristics

Using the ideal gas law as the fundamental framework, the proportional relationship between the pressure, density, and temperature of underground airflow was analyzed. By introducing time as a fundamental variable, a density-pressure-temperature coupling equation that describes the variation patterns of airflow state properties was proposed. This results in a

transient heat and airflow flow characteristics model in time series. Using this characteristics model as a correction variable, fluctuation equations for natural ventilation pressure and ventilation resistance over time were further proposed, forming a model of fluctuating natural ventilation pressure and ventilation resistance in time series. Ultimately, an airflow regulation model based on the heat and airflow coupling characteristics of the airflow was constructed.

## 2.1. Transient heat and airflow flow characteristics model of airflow in time series

Assuming the underground airflow is moist air with a relative humidity of $\varphi$, derive the calculation equation for airflow pressure and density based on Dalton's law [22]:

$$\rho = \rho_{\mathrm{dry}} + \rho_{\mathrm{humid}} = \left(1/T\right)\left[\left(\left(P - \varphi P_{\mathrm{sa}}\right)/R_{\mathrm{dry}}\right) + \left(\varphi P_{\mathrm{sa}}/R_{\mathrm{humid}}\right)\right] \tag{1}$$

Here, $P$ is the airflow pressure, Pa; $\rho$ is the airflow density, kg/m³; $\rho_{\mathrm{dry}}$ is the dry air density, kg/m³; $\rho_{\mathrm{humid}}$ is the water vapor density, kg/m³; $T$ is the airflow temperature, K; $P_{\mathrm{sa}}$ is the absolute partial pressure of saturated water vapor, Pa; $R_{\mathrm{dry}}$ and $R_{\mathrm{humid}}$ are the gas constants for dry air and water vapor, with values $R_{\mathrm{dry}} = 287.04$ and $R_{\mathrm{humid}} = 461.39$, J/(kg·K).

$P_{\mathrm{sa}}$ represents the water vapor pressure in the airflow when the water vapor reaches dynamic saturation. As the airflow temperature $T$ increases, the amount of water vapor that the air can hold increases, leading to a higher $P_{\mathrm{sa}}$; conversely, when $T$ decreases, $P_{\mathrm{sa}}$ decreases. The functional relationship between $P_{\mathrm{sa}}$ and $T$ can be described by the Goff-Gratch equation (2).

$$P_{\mathrm{sa}}(T) = 10^{\left(10.80\left(1 - T_0/T\right) - 5.03 \times \lg\left(T/T_0\right) + \left(1.50 \times 10^{-4}\right)\left[1 - 10^{-8.30 \times \left(T/T_0 - 1\right)}\right] + \left(4.29 \times 10^{-4}\right)\left[10^{4.77 \times \left(1 - T_0/T\right)} - 1\right] + 0.79\right)} \tag{2}$$

Here, $T_0$ is the triple point temperature of water, with a value of 273.16 K;

In a non-steady-state atmospheric environment, the airflow temperature at the inlet of the shaft is the atmospheric temperature that dynamically changes with time $t$ (unit: h). When the airflow enters the underground mine, it exchanges heat with the underground heat environment, causing the airflow temperature $T$ to change dynamically over time. Additionally, under the influence of the gradient flow field, the self-compression heat generated by the change in gravitational potential energy of the airflow leads to changes in volume forces and along-the-way resistance losses. As a result, in the spatial scale, the airflow pressure $P$ changes dynamically with $T$ over time. Therefore, in the spatial scale, the airflow temperature $T$, pressure $P$, and the partial pressure of saturated water vapor $P_{\mathrm{sa}}$ are all functions of time $t$: $T(x, y, z) = T(x, y, z, t)$, $P(x, y, z) = P(x, y, z, t)$, $P_{\mathrm{sa}}(x, y, z) = P_{\mathrm{sa}}(T) = P_{\mathrm{sa}}(x, y, z, t)$. By combining equation(1) and (2), the density-pressure-temperature coupling equation(3) can be obtained, which describes the changes of airflow state properties in the spatial scale in time series: $\rho = f_1(P, T, t)$:

$$\rho = \left(1/T(t)\right)\left[\left(\left(P(t) - \varphi P_{\mathrm{sa}}(t)\right)/R_{\mathrm{dry}}\right) + \left(\varphi P_{\mathrm{sa}}(t)/R_{\mathrm{humid}}\right)\right] = \left(3.48 \times 10^{-3}\left(P(t) - 0.38\varphi P_{\mathrm{sa}}(t)\right)\right)/T(t) \tag{3}$$

To quantitatively describe the degree to which the airflow in the underground mine deviates from an isentropic or isoheat process under different conditions, and to reflect the non-ideal characteristics of the actual underground thermodynamic process, the polytropic exponent $n$ is defined as a characteristic parameter describing the thermodynamic process of the airflow. Based on this, the state change law of the polytropic process is extended to the following polytropic exponent-density-pressure coupling equation(4): $n = f_2(P, \rho, t)$.

$$P(1/\rho)^n = \text{const} = P_0(1/\rho_0)^n \rightarrow n = (\ln P(t) - \ln P_0)/(\ln \rho(t) - \ln \rho_0) \qquad (4)$$

By using numerical solutions, the equations $\rho = f_1(P, T, t)$ and $n = f_2(P, \rho, t)$ are further transformed into a comprehensive explicit expression $\rho = f_3(n, P, T, t)$, forming a transient heat and airflow flow characteristics model that describes the change law of airflow state properties in time series, applicable to actual mines, as shown in equation (5) below.

$$\begin{cases} \rho - f_1(P,T,t) = 0 \\ n - f_2(P,\rho,t) = 0 \end{cases} \rightarrow \rho = f_3(n,P,T,t) \qquad (5)$$

From equation(5), it can be seen that within the framework of the airflow in the mine being a polytropic process formed by the coupling of aerodynamic and thermodynamic processes, the density of the airflow with heat and airflow coupling characteristics is simultaneously influenced by temperature and pressure. In time series, this exhibits dynamic fluctuation characteristics with the polytropic exponent $n$ as a thermodynamic parameter, forming a density-pressure-temperature coupling in the form $\rho = f_3(n, P, T, t)$ (after simplified as $\rho(t)$).

## 2.2. Fluctuating natural ventilation pressure and ventilation resistance model in time series

(1) **Correction of fluctuating natural ventilation pressure.** In traditional mine main fan parameter planning activities, natural ventilation pressure is usually simplified and treated as a static linear equation. However, under the heat and airflow coupling characteristics of airflow, the dynamically changing state properties of the airflow cause natural ventilation pressure to no longer follow a linear pattern. To describe the dynamic fluctuation characteristics of natural ventilation pressure in time series, the transient heat and airflow flow characteristics model $\rho(t)$ is introduced to correct the natural ventilation pressure $h_N$ (unit: Pa). Considering the descending airflow direction as the positive direction, the time series fluctuation equation for natural ventilation pressure is proposed in equation (6):

$$h_N(t)' = \sum(h_{Ni} - h_{No}) = \sum_m\left(\left(\sum_j P_{in}(t) + \rho_j(t)gZ_j\right) - \left(\sum_k P_{out}(t) + \rho_k(t)gZ_k\right)\right) \qquad (6)$$

Here, $h_N(t)'$ is the corrected fluctuating natural ventilation pressure, Pa; $h_{Ni}$ is the gravitational pressure of the vertical descending air column, Pa; $h_{No}$ is the gravitational pressure of the vertical ascending air column, Pa; $m$ is the number of control ventilation paths with different end node heights; $P_{in}(t)$ is the starting pressure value for the descending airflow, Pa; $P_{out}(t)$ is the starting pressure value for the ascending airflow, Pa; $Z$ is the vertical height difference of the roadway, m; $j$ is the identifier for the descending airflow path, $j \in \{1, 2, 3, …, i\}$; $k$ is the identifier for the ascending airflow path, $k \in \{1, 2, 3, …, i\}$; $g$ is the gravitational acceleration, m/s².

(2) **Correction of fluctuating ventilation resistance.** Based on the law of turbulent frictional resistance, the calculation equation for ventilation resistance in mine roadways is given in equation (7) [23]. From equation (7), it can be seen that for a standardized mine roadway, $R_i$ is proportional to the friction coefficient $\alpha$, Since $\lambda$ in a fully turbulent state is only related to the relative roughness of the wall surface, the $\lambda$ value for a standardized mine roadway is constant, making $\alpha$ proportional only to the airflow density $\rho$. Thus, solving for the fluctuating ventilation resistance in mine roadways essentially involves using $\rho(t)$ to determine the dynamically fluctuating $\alpha$.

$$h_{i-r} = R_i q_i^2 = (\alpha_i L_i U_i / S_i^3)q_i^2 = (\lambda_i \rho_i L_i U_i / 8S_i^3)q_i^2 \qquad (7)$$

Here, $h_{i\text{-r}}$ is the ventilation resistance of roadway $i$, Pa; $R_i$ is the frictional resistance of roadway $i$, N·s²/m⁸; $q_i$ is the airflow volume passing through roadway $i$, m3/s; $\alpha_i$ is the friction coefficient of roadway $i$, kg·m³; $L_i$ is the length of roadway $i$, m; $U_i$ is the perimeter of the cross-section of roadway $i$,m; $S_i$ is the cross-sectional area of roadway $i$,m²; $\lambda_i$ is the experimental proportional coefficient of roadway $i$; $\rho_i$ is the density of the airflow passing through roadway $i$, kg/m³.

Due to the difficulty of measuring wall smoothness in practical engineering, the determination of the friction coefficient $\alpha$ in mine roadways typically relies on measuring the air pressure and air volume in the roadway using barometers and anemometers, and then performing data inversion based on equation (7). Alternatively, the standard $\alpha$ value can be directly obtained from tables for an airflow density $\rho$ of 1.2 kg/m³. These methods, however, neglect the transient changes in the state properties of airflow under the heat and airflow coupling characteristics, resulting in $\alpha$ values that can only be used to solve for the resistance of the roadway at a specific moment. Using a single moment's roadway resistance value to replace the dynamically changing resistance value in time series cannot accurately reflect the operation characteristics of the ventilation network in real-time. Based on the definition of $\alpha$, the time-series fluctuation equation for ventilation resistance is derived in equation (8).

$$h_{i-\text{r}}(t)' = \left(\alpha_i' L_i U_i \big/ S_i^3\right) q_i^2 = \left(\left(\rho_i(t)/\rho_i\right)\alpha_i L_i U_i \big/ S_i^3\right) q_i^2 = \left(\rho_i(t)/\rho_i\right) R_i = R_i(t)' q_i^2 \qquad (8)$$

Here, $h_{i\text{-r}}(t)'$ is corrected fluctuating ventilation resistance of roadway $i$, Pa;.$\alpha_i'$ is fluctuating frictional coefficient of roadway $i$, kg·m³;.$\rho_i(t)$ is time-dependent airflow state parameter (density) of roadway $i$, kg/m³; $R_i(t)'$ is fluctuating frictional resistance of roadway $i$, N·s²/m⁸;

**(3) Quantifying the impact of fluctuation corrections on the ventilation system.** Since airflow in the ventilation system exhibits the characteristics of a continuous medium, applying fluctuation corrections to the natural ventilation pressure and ventilation resistance of individual roadways in time series will inevitably cause a chain reaction in related roadways (or fans) influenced by these corrections, leading to fluctuations in the air volume throughout the entire ventilation network. To synchronously map the dynamic continuous operational state of the ventilation system, the corrected fluctuating values $h_N(t)'$ and $R_i(t)'$ are used as initial iteration values based on the law of pressure balance [23]. Using continuous time as the iteration count, a time-series ventilation network calculation model is constructed (as shown in Equation (9)). The cross algorithm is then used to perform reverse iterations to calculate the air volume $q_i(t)$ (unit: m³/s) in each branch of the ventilation network in time series. By accurately calculating the network air volume changes $q_i(t)'$ caused by the corrections in ventilation resistance and natural ventilation pressure in time series, and the feedback effects of $q_i(t)'$ changes on ventilation resistance and natural ventilation pressure, the precise description of the dynamic behavior of the entire ventilation system is achieved.

$$\begin{cases} q(t)' = C^T q_y(t)' \\ CR(t)'_{\text{diag}} \left| C^T q_y(t)' \right|_{\text{diag}} C^T q_y(t)' - CH_f - H_N(t)' = 0 \end{cases} \qquad (9)$$

Here, $q(t)'$ is the branch airflow volume vector; $C$ is the fundamental loop matrix; $q_y(t)'$ is the co-tree branch airflow volume vector; $R(t)'$ is the corrected branch frictional resistance column vector; diag indicates a diagonal matrix formed by a vector; $H_f$ is the fan characteristics column vector; $H_N(t)'$ is the corrected fluctuating natural ventilation pressure column vector.

In summary, the fluctuating natural ventilation pressure and ventilation resistance model in time series was constructed as shown in Equation (10).

$$
\begin{cases}
h_{\mathrm{N}}(t)' = \sum_{m}\left(\left(\sum_{j} p_{\mathrm{in}}(t) + \rho_{j}(t)gZ_{j}\right) - \left(\sum_{k} p_{\mathrm{out}}(t) + \rho_{k}(t)gZ_{k}\right)\right) \\
h_{i-\mathrm{r}}(t)' = \left(\alpha_{i}'L_{i}U_{i}/S_{i}^{3}\right)q_{i}^{2} = \left(\left(\rho_{i}(t)/\rho_{i}\right)\alpha_{i}L_{i}U_{i}/S_{i}^{3}\right)q_{i}^{2} = \left(\rho_{i}(t)/\rho_{i}\right)R_{i} = R_{i}(t)' q_{i}^{2} \\
q(t)' = \mathrm{C}^{\mathrm{T}}q_{y}(t)' \\
\mathrm{CR}(t)'_{\mathrm{diag}}\left|\mathrm{C}^{\mathrm{T}}q_{y}(t)'\right|_{\mathrm{diag}}\mathrm{C}^{\mathrm{T}}q_{y}(t)' - \mathrm{CH}_{\mathrm{f}} - \mathrm{H}_{\mathrm{N}}(t)' = 0
\end{cases} \tag{10}
$$

## 2.3. Airflow regulation model in time series

**(1) Objective function.** The fluctuating natural ventilation pressure and ventilation resistance was used as the environmental variable, the adjustment resistance was used as the decision variable, and the minimum power consumption of the ventilation system under the premise of ensuring the required air volume in the underground branch was used as the airflow control objective. The objective function was constructed as follows:

$$
\min \sum_{i=1}^{v} h_{i}(t)q_{i}(t)' = \min \sum_{i=1}^{v}\left(\left(h_{i-\mathrm{r}}(t)' + \Delta h_{i}(t) + h_{\mathrm{N}}(t)'\right)\left(q_{i}(t)'\right)\right) \tag{11}
$$

Here, in time series, $h_{i}(t)$ is the ventilation pressure in roadway $i$, Pa; $\Delta h_{i}(t)$ is the adjustment resistance in roadway $i$; $v$ is the number of mine roadways.

**(2) Constraint conditions.**

i) Mass flow conservation law [24]

Despite the fact that the state properties values of the airflow and ventilation parameters change interactively over time, the dynamic changes in roadway airflow must still satisfy the fundamental thermodynamic constraint of the conservation of airflow. Considering the dynamic changes in the state properties of the airflow, the mass flow balance law is used as the system constraint condition, such that the mass of the airflow flowing into a node per unit time is equal to the mass of the airflow flowing out of the node. The variable function expression is:

$$
BG = 0 \tag{12}
$$

Here, $B$ is the fundamental incidence matrix; $G$ is the branch mass flow vector.

ii) Ventilation pressure conservation law [24]

$$
\mathrm{C}\left(\mathrm{H}_{\mathrm{f}} + \mathrm{H}_{\mathrm{N}}(t)' + \mathrm{H}_{\mathrm{r}}(t)'\right) = 0 \tag{13}
$$

Here, $H_{\mathrm{r}}(t)'$ is the column vector of the corrected fluctuating ventilation resistance in the branches.

iii) Ventilation pressure iterative correction

The range delineated by the maximum and minimum blade angle characteristic curves, the 60% efficiency curve, and the 90% maximum working ventilation pressure curve is defined as

the reliable working zone for the fan, as shown in Fig 1. When the working condition point falls within this zone, the fan operates reliably. To avoid convergence of this point toward an incorrect root ($Q_{f2}$) during the solution of the airflow control model due to the iterative calculation of the fan ventilation pressure, the ventilation pressure outside the reliable working zone was set as the boundary pressure of the zone. The corrected fan characteristic equation is as follows:

$$h_f = \begin{cases} a_0 + a_1 Q + a_2 Q^2, Q \geq Q_1 \\ h_1, Q < Q_1 \end{cases} \tag{14}$$

Here, $Q$ is the working air volume of the fan, m3/s; $Q_1$ is the boundary air volume of the reliable working range of the fan, m3/s.

In summary, the mathematical model of the airflow regulation can be expressed as follows:

$$\begin{cases} \min \sum_{i=1}^{v} h_i(t) q_i(t)' = \min \sum_{i=1}^{v} \left[ \left( h_{i-r}(t)' + \Delta h_i(t) + h_N(t)' \right) \left( q_i(t)' \right) \right] \\ s.t. \\ \mathrm{BG} = \boldsymbol{0} \\ \mathrm{C} \left( \mathrm{H_f} + \mathrm{H_N}(t)' + \mathrm{H_r}(t)' \right) = \boldsymbol{0} \\ h_f = \begin{cases} a_0 + a_1 Q + a_2 Q^2, Q \geq Q_1 \\ h_1, Q < Q_1 \end{cases} \end{cases} \tag{15}$$

The airflow regulation model integrating the transient heat and airflow flow characteristics model of airflow and the fluctuating natural ventilation pressure and ventilation resistance model, constructed based on heat and airflow coupling characteristics, enables precise characterization of time-dependent airflow state variations. This system demonstrates real-time environmental responsiveness and dynamic airflow state adjustment capabilities, achieving optimized air volume distribution that effectively fulfills the adaptive control requirements of mine ventilation systems in complex dynamic environments.

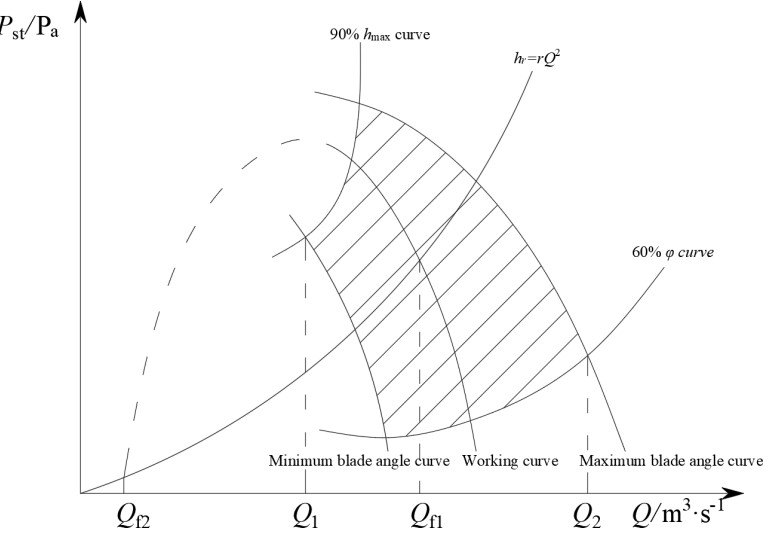

**Fig 1. Reliable working area for fans.**

## 2.4. Solution of the airflow regulation model

From the perspective of mathematical logic, the core of solving the objective function of the constructed airflow regulation model, which is based on the heat and airflow coupling characteristics, lies in analyzing the interaction between the dynamic environmental variables and the decision variable. Based on the interaction mechanism between the variables, the objective function, $\min N = \min N_1 + \min N_2$, is divided into a multidimensional function $N_1$ and a linear programming model function $N_2$ using the fractional step method (as shown in Equation (16)). The decomposed $N_1$ primarily focuses on analyzing the impact of fluctuating natural ventilation pressure and ventilation resistance on the power consumption of the ventilation system, reflecting the constraints imposed by dynamic underground environmental conditions on airflow. On the other hand, $N_2$ focuses on the optimization process of adjustment resistance, highlighting the role of ventilation system design parameters in regulating power consumption. By decomposing the objective function, the independent optimization of environmental variables and decision variables is achieved, directly reducing the complexity of solving the model.

$$\min N_1 = \min \sum_{i=1}^{v} \left( \left( h_{i-\mathrm{r}}(t)' + h_{\mathrm{N}}(t)' \right) \left( q_i(t)' \right) \right),$$

$$\min N_2 = \min \sum_{i=1}^{v} \left( \Delta h_i(t) q_i(t)' \right) \tag{16}$$

The basic solution criterion for $N_1$ is to calculate the instantaneous power consumption of the dynamically changing ventilation system at each time step, aiming to minimize the total system power consumption by optimizing the instantaneous consumption values. Specifically, solving $N_1$ involves dynamic modeling of the airflow state in time series, ensuring that changes in natural ventilation pressure, ventilation resistance, and airflow at each time step comply with the law of conservation of mass and other physical constraints. Numerical methods are used to find the optimal power consumption of the system by optimizing the airflow distribution at each time step to achieve minimal power consumption.

The focus of solving $N_2$ is on optimizing the impact of adjustment resistance on the power consumption of the ventilation system. By applying the simplex method to solve the linear programming function, multiple sets of $N_2$ solutions, solution sets for different time steps are obtained, with each set representing the optimal adjustable resistance branch set $\{\Delta h_i\}$ at the current time step. By gradually adjusting the resistance in real-time, the overall power consumption of the system is further reduced while meeting the airflow requirements of each branch underground.

**Step 1: Constructing the ventilation network model.  i) Building the network topology:** Based on the data structure between branches and nodes in the ventilation network, construct the set of branches and the set of nodes, and create the undirected graph of the ventilation network. Construct a zero matrix, using branch resistance and branch air volume as weights, and assign values to the matrix by combining the set of branches and the set of nodes. This completes the initial setup of the branch resistance set and the air volume set for the ventilation network. Construct the column vector for the fan characteristics based on the fan characteristic equations.

**ii) Minimum spanning tree calculation:** Initialize parameters and use Kruskal algorithm to search for the minimum weight edges in the network, completing the calculation of the minimum spanning tree and the co-tree.

**iii) Solving the fundamental loop matrix $C$:** Calculate the fundamental incidence matrix $B$ of the ventilation network based on the relationships between nodes and branches; using the equation $BC^{\mathrm{T}}=0$, solve for the fundamental loop matrix $C$ of the network.

**Step 2: Constructing the main function of the airflow regulation model.** **i) Initial condition setup:** Set the decision variable independent optimization function $N_2$ as the main function and the environmental variable independent optimization function $N_1$ as the auxiliary function. Set the maximum number of iterations $T_{\mathrm{iter}}$. Set the iteration termination condition as when the correction value of the main optimization variable $\Delta h_i$ is less than the iteration precision $\sigma$. keep the main function model in its default state.

**ii) Spatiotemporal discretization:** Use the stepwise difference method to discretize the mine roadways into finite micro-segments; use the environmental parameters at the end of the $i$-th segment as the starting environmental parameters for the $(i+1)$-th segment. Set the time variable $t$ with an interval of 1 hour, ranging from 0 to 23, resulting in a total time interval of $[0, 23]$ hours, discretized into time steps 1, 2, …, $t_u$, …, 23.

**iii) Function construction:** Construct the main function $N_2$ and the auxiliary solving function $N_1$ based on the objective function decomposition equation (16). Obtain the branch density data $\rho(t_u)$ at each time step through model1. Calculate the fluctuating natural ventilation pressure $h_{\mathrm{N}}(t_u)'$ and the ventilation resistance $h_{i\text{-}\mathrm{r}}(t_u)'$ through model2. Determine the optimal solution for the airflow regulation model through model3.

**iv) Function solving and output:** Complete the optimal solution for airflow regulation and output the data for the adjustment branch set.

**Step 3: Define the model1 for transient heat and airflow flow characteristics of airflow in time series.** **i) Fitting of state properties values for external mine nodes:** Input atmospheric environmental field parameters to construct temperature, pressure, and density vectors. Utilize the polyfit function to complete polynomial fitting with $t$ as the independent variable, obtaining $T_1(t)$、 $P_1(t)$ and $\rho_1(t)$.

**ii) Construction of computational model for internal mine node state properties:** Assign values to density, heat release coefficient, and heat conductivity matrices based on rock formation properties, and calculate the unsteady heat transfer coefficient. Establish a corresponding transient heat and airflow flow characteristics model for airflow according to the polytropic exponent-density-pressure-temperature coupling equation: $\rho=f_3(n, P, T, t)$.

**iii) Calculation of state properties values:** Set dynamically changing atmospheric node state parameters in time series as initial node values. Implement segmented recursive calculation of the transient model under discrete spatial scales.

**Step 4: Define the model2 for fluctuating natural ventilation pressure and ventilation resistance in time series.** **i) Updating ventilation resistance weights:** Reallocate the global ventilation network resistance weights based on the relationship $R_{t}+_1=R_t\times\rho_{t}+_1/\rho_t$.

**ii) Calculation of fluctuating natural ventilation pressure and ventilation resistance:** Substitute $\rho_{t}+_1$ and $R+_1$ as initial values into the fluctuating natural ventilation pressure and ventilation resistance model. Compute the transient fluctuation values of natural ventilation pressure and ventilation resistance in time series, and output the natural ventilation pressure $h_{\mathrm{N}}(t_u)'$ and ventilation resistance $h_{i\text{-}\mathrm{r}}(t_u)'$ data at each time step.

**iii) Optimizing airflow distribution:** Use $h_{i\text{-}\mathrm{r}}(t_u)'$ and $h_{\mathrm{N}}(t_u)'$ at each time step as initial values in the ventilation network calculation model. Complete the calculation of the global airflow values at each moment, and output the airflow data $q_i(t_u)'$ at each time step.

**Step 5: Solving the airflow regulation model3.** **i) Solving the main function:** Transform the objective function and constraints into the standard form of linear programming. Construct the initial simplex tableau and check if the reduced costs meet the non-negativity condition to determine if the current solution is optimal. If the reduced costs are

non-negative, end the iteration and output the current solution. If any reduced cost is negative, select the column corresponding to the most negative reduced cost to enter the basis. Calculate the ratios to determine the variable to leave the basis, and update all variable values centered around the pivot using Gaussian elimination to construct a new simplex tableau. Repeat the non-negativity check and variable replacement steps until the iteration conditions are met. Extract the optimal solution of the main function and the corresponding decision variable values from the final simplex tableau.

**ii) Solving the auxiliary function:** Update the ventilation resistance matrix based on the solution of the main function. Perform a secondary calculation of the ventilation network, generating airflow $q_i(t_u)''$ and ventilation resistance $h_{i\text{-}r}(t_u)''$ data for each time step after resistance adjustment. Use the corresponding airflow $q_i(t_u)''$, natural ventilation pressure $h_N(t_u)'$, and ventilation resistance $h_{i\text{-}r}(t_u)''$ as data sources at each time step. Calculate the instantaneous power consumption at each time step. Analyze the consumption trend of the system in time series based on the calculated $N_1$ value. Solve for the minimum value min-$N_1$ to complete the solution of the auxiliary function.

## 3. Practical feasibility validation of proposed model

The premise for establishing the airflow regulation model based on the heat and airflow coupling characteristics of airflow is that the underground airflow is a variable flow driven by the coupling of aerodynamic and heat effects. To verify whether the proposed regulation model and the algorithm used to solve the model can be applied in actual production mines, using the dynamic changes in the state properties of variable airflow under spatiotemporal characteristics as the validation mechanism, the Gucheng Coal Mine located in Changzhi City, China, was taken as an engineering subject. Using the unsteady atmospheric environment field, underground heat environment field, and gradient flow field as boundary conditions, and the actual surveyed mine environment information as initial conditions, a numerical model of unsteady heat transfer for Gucheng Coal Mine is constructed. Using the CFD simulation technology utilized to solve the model, obtaining the state properties of the airflow in time series. These property solutions are used as references to verify the heat and airflow coupling characteristics and the reliability of the proposed transient heat and airflow flow characteristic model, and the proposed airflow regulation model and algorithm were found to be applicable to actual mines.

### 3.1. Mathematical Model Setup

To concentrate on the study of heat and airflow coupling characteristics, the following assumptions are made:

(1) Considering the small variation in humidity in individual mine roadways, convective mass transfer is neglected, the water phase remains unchanged, and there is no latent heat of vaporization.

(2) The airflow is treated as a continuous, compressible turbulent flow. During the flow process, except for changes in state properties, other properties remain fixed.

(3) The model boundaries are assumed to be non-slip, non-deformable, and without air leakage.

(4) The surrounding rock is assumed to be homogeneous and isotropic sandstone.

Based on these assumptions, the analysis of the airflow in an unsteady environmental field and gradient flow field is transformed into an analysis of the internal energy conversion

process of compressible airflow exchanging heat with the solid boundaries of the model. According to Fourier's law and Newton's law of cooling [25], the analysis of internal energy changes due to heat exchange is converted into the numerical solution of airflow temperature, pressure, and density.

The momentum, energy, and mass conservation equations are combined to construct the heat transfer balance equation, non-isoheat flow equation, and volumetric force equation [25], forming the mathematical model of heat and airflow. Solving this model will yield the transient temperature field $T(x, y, z, t)$, pressure field $P(x, y, z, t)$, and density field $\rho(x, y, z, t)$. Based on the Reynolds-averaged method, the heat and airflow model is established as follows:

$$\begin{cases} \rho S C_p\, \partial T/\partial t + \rho S C_p \mathbf{u} \mathbf{e} \cdot \nabla T = \nabla \cdot \left(S\lambda_k \nabla T\right) + \left(1/2\right) f_D \left(\rho S/d_h\right)|\mathbf{u}|\mathbf{u}^2 + A + (hD)_{eff}\left(T_{ext} - T\right) \\ \rho \partial \mathbf{u}/\partial t = -\nabla p \cdot \mathbf{e} - \left(1/2\right) f_D \left(\rho/d_h\right)|\mathbf{u}|\mathbf{u} + \mathbf{F} \times \mathbf{e} \\ \partial S\rho/\partial t + \nabla\left(S\rho \mathbf{u}\mathbf{e}\right) = 0 \\ \nabla\left(\rho \overline{\mathbf{u}}\overline{\mathbf{u}}\right) = \mathbf{F} - \nabla\left(\overline{P}\right) - \partial\left(\rho \overline{\mathbf{u}}\right)/\partial t + \nabla\left(\underset{=}{\tau} + \underset{=}{\tau'}\right) \\ \partial\left(\rho \varepsilon\right)/\partial t + \nabla\left(\rho \varepsilon \mathbf{u}\right) = \nabla\left(\mu\left(\nabla \mathbf{u} + (\nabla \mathbf{u})^T\right) - 2/3\mu\nabla \mathbf{u}\right) + \varepsilon_k - \varepsilon_e + \varepsilon_p \end{cases}$$

Here, $C_p$ is specific heat at constant pressure, J/(kg·K); $\mathbf{e}$ is tangential vector; $\mathbf{u}$ is velocity vector, m/s; $\lambda_k$ is thermal conductivity of the surrounding rock, W/(m·K); $f_D$ is Darcy friction factor; $d_h$ is hydraulic diameter of the model pipe, m; $A$ is heat source, W·m; $h$ is heat transfer coefficient, W/(m²·K); $D$ is wetted perimeter of the pipe, m; $T_{ext}$ is temperature on the outer side of the pipe wall, K; $\mathbf{F}$ is volumetric force, N; $\mu$ is dynamic viscosity, Pa·s; $\tau$ is surface stress, N; $\overline{\mathbf{u}}$ is mean velocity, m/s; $\tau'$ is additional Reynolds stress caused by airflow pulsation, N; $\varepsilon$ is turbulent dissipation rate, m²·s³; $\varepsilon_k$ is dissipation of turbulent kinetic energy; $\varepsilon_e$ is momentum dissipation caused by pressure gradient; $\varepsilon_p$ is additional momentum dissipation caused by pulsating velocity.

In an actual mine, when airflow moves through the roadways, friction with the wall surfaces generates resistance. Therefore, it is necessary to account for the pressure loss due to airflow in a rough pipe model. Using the Darcy-Weisbach equation, by setting the pressure drop value equal to the actual ventilation resistance value, the equation for the pressure loss along the length of airflow in a rough pipe model is established (Equation (18)). The relationship between the Darcy friction factor $f_D$ and the Reynolds number is used to calculate the wall roughness $\eta$ of the model (calculation equation shown in Equation (19)).

$$\Delta p = f_D \left(L/d_h\right)\left(\rho(\mathbf{u}\cdot\mathbf{e})^2/2\right) = \left(\lambda\rho LU/8S^3\right)Q^2 = \lambda\left(L/d_h\right)\left(\rho(\mathbf{u}\cdot\mathbf{e})^2/2\right) \tag{18}$$

$$f_D = 8\left(\left(8\mu/\rho u d_h\right)^{12} + \left(\left(-2.46\ln\left(\left(7\mu/\rho u d_h\right)^{0.9} + 0.27\left(\eta/d_h\right)\right)\right)^{16} + \left(37530\mu/\rho u d_h\right)^{16}\right)^{-1.5}\right)^{1/12} \tag{19}$$

Here, $\Delta p$ is pressure drop, Pa; $\eta$ is wall surface roughness, m.

## 3.2. Full-scale modeling of an equivalent ventilation system

A "three in, two out" partition ventilation system has been adopted for the Gucheng Coal Mine, where the main inclined shaft and auxiliary vertical shaft serve as inlet airflow paths, and the central outlet vertical shaft serves as the outlet airflow path in the central area, the Taoyuan inlet vertical shaft serves as the inlet airflow path, and the Taoyuan outlet vertical shaft serves as the outlet airflow path in Taoyuan area. Fig 2 shows a schematic of the ventilation system control of Gucheng Coal Mine, drawn based on the relationship between the fan and the mining area network.

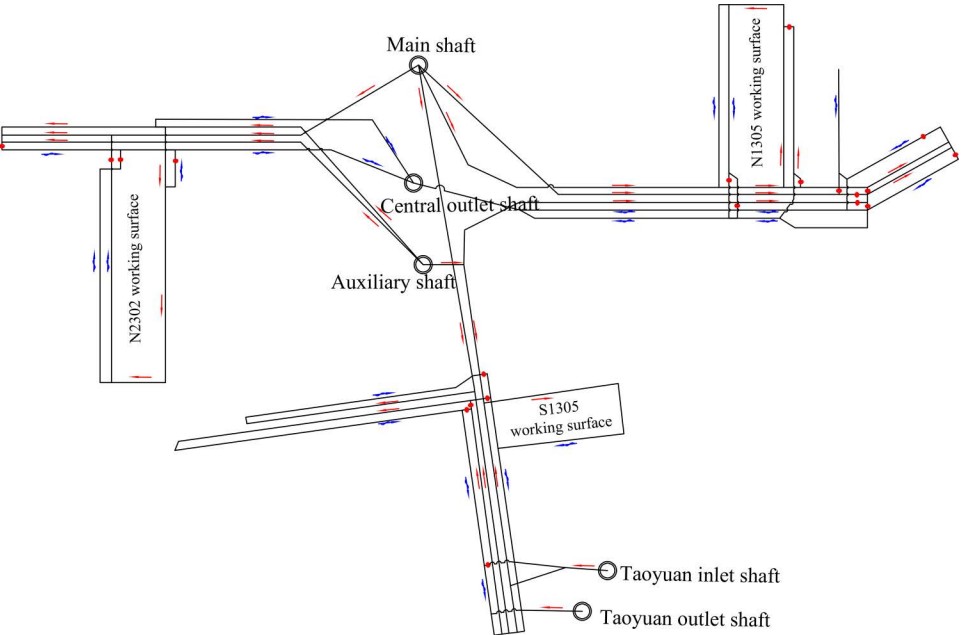

**Fig 2. Mining area network controlled separately for each fan.**

The central inlet shaft (main shaft and auxiliary shaft) primarily supply air to the eastern, western, and some southern areas of the mine, while the Taoyuan inlet shaft mainly supplies air to the southern area. Consequently, the ventilation system of the Gucheng Coal Mine can be equivalently represented by a near H-type structural system, as shown in Fig 3. In this representation, $e_1$ represents the central inlet shaft, $e_2$ serves as the inlet airpath for the eastern and western areas of the mine from the central inlet shaft, $e_3$ represents the Taoyuan inlet shaft, $e_4$ connects the Taoyuan inlet shaft to the southern inlet airpath, $e_5$ connects the central inlet shaft to the southern inlet airpath, $e_6$ connects the outlet airpath from the eastern and western areas to the central outlet shaft, $e_7$ is the southern outlet airpath, $e_8$ is the central outlet shaft, and $e_9$ is the Taoyuan outlet shaft.

Based on Fig 3 and geological survey data, an unsteady heat transfer numerical model of Gucheng Coal Mine was established. Since most of the main roadways underground are coal roadways, the wall boundary material of the shaft is set to sandstone during modeling, while the boundary material of the other branches is set to lean coal. The initial physical conditions of the model are shown in Table 1.

### 3.3. Setting of boundary conditions

#### (1) Heat transfer boundary condition.

i) Atmospheric environment temperature field

The environmental parameters measured during the ventilation resistance testing program in Changzhi City (the city where the Gucheng Mine is located) on June 22, 2023, were used as the unstable atmospheric environment field data, based on the least-squares discriminant criterion, the atmospheric temperature-time polynomial analytic function $T_1(t)$ and the atmospheric pressure-time polynomial analytical function $P_1(t)$ are constructed, forming the atmospheric temperature field $T_1(t)$ and the atmospheric pressure field $P_1(t)$ in time series. $T_1(t)$

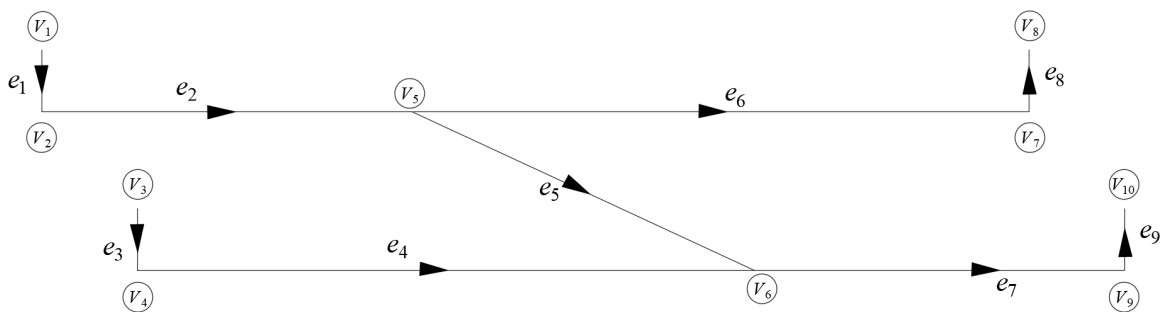

**Fig 3. Equivalent diagram of the ventilation system network of the Gucheng Coal Mine, where the various branches are denoted by the letter "*e*," and "*V*" represents the nodes.**

**Table 1. Initial physical conditions of model.**

| Equivalent branch | Function | Length $L$/m | Altitude difference $Z$/m | Equivalent area $S$/m² | Equivalent hydraulic diameter $d_h$/m | Thermal conductivity of wall layer $\lambda_k$/(W·m⁻¹·K⁻¹) | Surface roughness $\eta$/m |
|---|---|---|---|---|---|---|---|
| $e_1$ | the central inlet shaft | 500 | 500 | 34.11 | 6.59 | 2.75 | 7.02×10⁻¹ |
| $e_2$ | the inlet airpath for the eastern and western areas of the mine from the central inlet shaft | 3000 | 0 | 32.37 | 6.42 | 0.31 | 2.59×10⁻³ |
| $e_3$ | the Taoyuan inlet shaft | 500 | 500 | 29.51 | 6.13 | 2.75 | 9.40×10⁻¹ |
| $e_4$ | the Taoyuan inlet shaft to the southern inlet airpath | 5000 | 0 | 31.77 | 6.36 | 0.31 | 1.33×10⁻⁴ |
| $e_5$ | the central inlet shaft to the southern inlet airpath | 3910 | 0 | 28.18 | 5.99 | 0.31 | 6.40×10⁻⁴ |
| $e_6$ | the outlet airpath from the eastern and western areas to the central outlet shaft | 5000 | 0 | 29.42 | 4.81 | 0.31 | 1.07×10⁻³ |
| $e_7$ | the southern outlet airpath | 3000 | 0 | 33.08 | 5.10 | 0.31 | 1.66×10⁻³ |
| $e_8$ | the central outlet shaft | 500 | -500 | 64 | 8 | 2.75 | 1.03 |
| $e_9$ | the Taoyuan outlet shaft | 500 | -500 | 64 | 8 | 2.75 | 9.44×10⁻¹ |

is applied as the inflow boundary condition for the model. The distribution of the unsteady atmospheric environmental field is shown in Fig 4 below.

$$T_0(t) = -8.64\times10^{-6}t^6 + 6.84\times10^{-4}t^5 - 0.02t^4 + 0.24t^3 - 1.01t^2 + 0.84t + 294.23$$

$$P_0(t) = 9.53\times10^{-5}t^6 - 7.54\times10^{-3}t^5 + 0.22t^4 - 2.94t^3 + 17.11t^2 - 35.37t + 93328.38$$

ii) Underground environment temperature field

By measuring the temperature in underground mine roadways, the underground temperature field $T_2(x, y, z)$ is obtained and used as the temperature boundary condition. The distribution of the underground temperature field data is shown in Fig 5 below.

**(2) Flow boundary conditions.** To avoid ill-conditioned boundaries, the airflow inlet boundaries were set as pressure boundaries(connected to the atmospheric pressure field $P_1$(t)), while the outlet boundaries were designated as velocity boundaries (constant flow velocity rate to simulate the extraction effect of the exhaust fan), The normal flow velocity at the central outlet shaft is set to 10.05 m/s, and the flow velocity at the Taoyuan outlet shaft is set to 10.34 m/s.

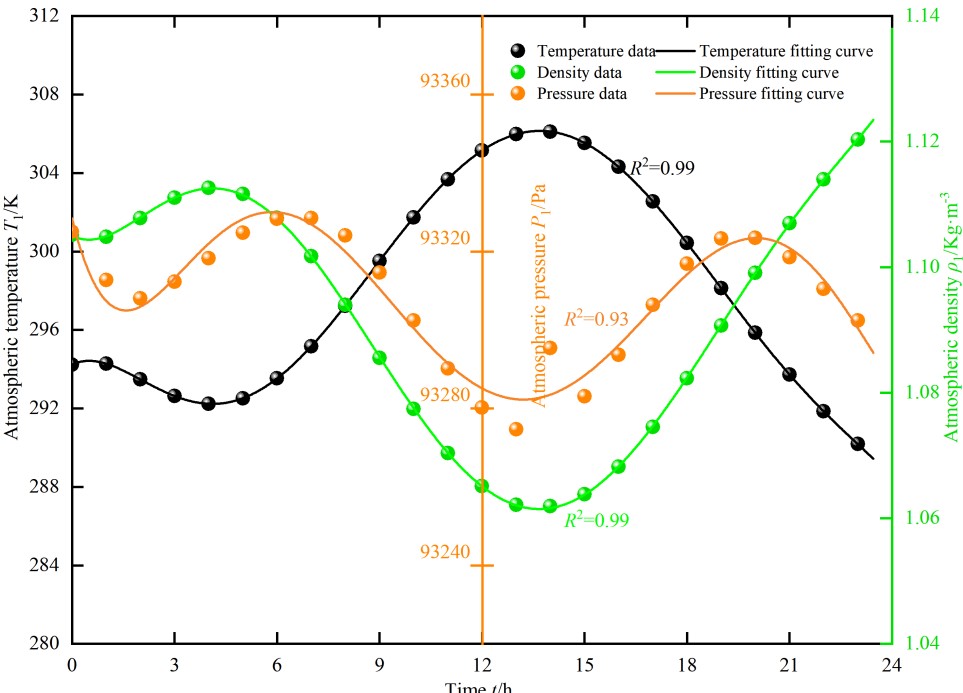

**Fig 4. Time-varying distribution of density, temperature, and pressure in the unsteady atmospheric environment.**

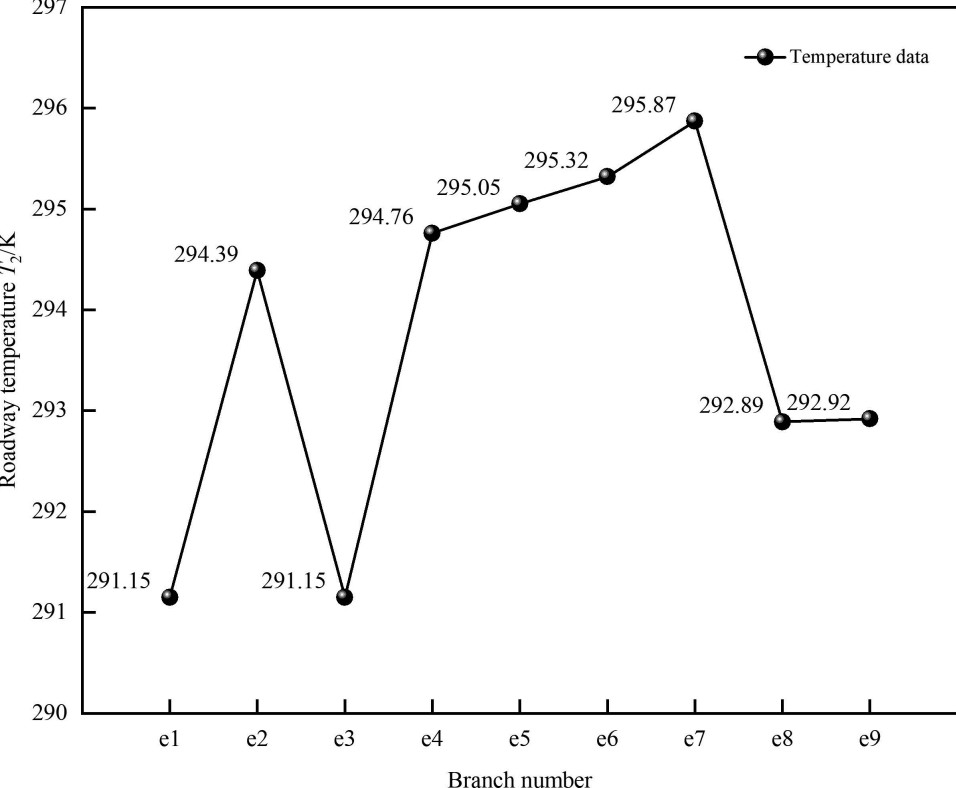

**Fig 5. Temperature distribution in mine roadways under the underground temperature field.**

### 3.4. Heat transfer model solution and simulation result analysis

Based on the boundary conditions, the model's heat transfer and flow restrictions were defined to complete the setup of the unsteady heat transfer numerical model for the Gucheng Coal Mine. The finite element method combined with an implicit algorithm was employed to discretize the model in both spatial and temporal dimensions. Using the mathematical model of heat flow as a foundation and constrained by the physical model and unsteady boundary conditions, an iterative convergence tolerance of 0.001 was set as the termination condition for calculations. This process involved constructing a set of spatially discretized finite element equations for each time step to solve for the transient state properties of the airflow.

The unsteady heat transfer model for the Gucheng Coal Mine was solved for $t \in [0, 23]$h, and the solution results for each time step were integrated to form a continuous time series. Data were sampled at the time series extremum points [0, 4, 14, 23]h to generate dynamic airflow temperature distribution map under spatiotemporal characteristics (Fig 6) (All subsequent results presented were sampled at these time series extremum points because these moments exhibit significant temperature change rates, with notable internal heat exchange and flow processes within the system. This approach represents the system's typical states at different times and avoids unnecessary complexity from long time series data).

Based on Fig 6, it can be observed that due to the external connection of the inlet shaft with the unsteady atmospheric environment, the airflow temperature at the shaft entrance fluctuates with the atmospheric temperature field $T_1$, exhibiting dynamic fluctuation characteristics. After the airflow enters the inlet shafts $e_1$, $e_3$, the temperature difference between the initial temperature of the mine roadways and the airflow causes heat transfer between the wall surface and the airflow, forming a heat flow gradient and resulting in dynamic changes in the internal temperature of the airflow. In the initial section of the inlet shaft, the airflow temperature is influenced by both $T_1$ and $T_2$, with $T_1$ being the dominant factor. As the airflow

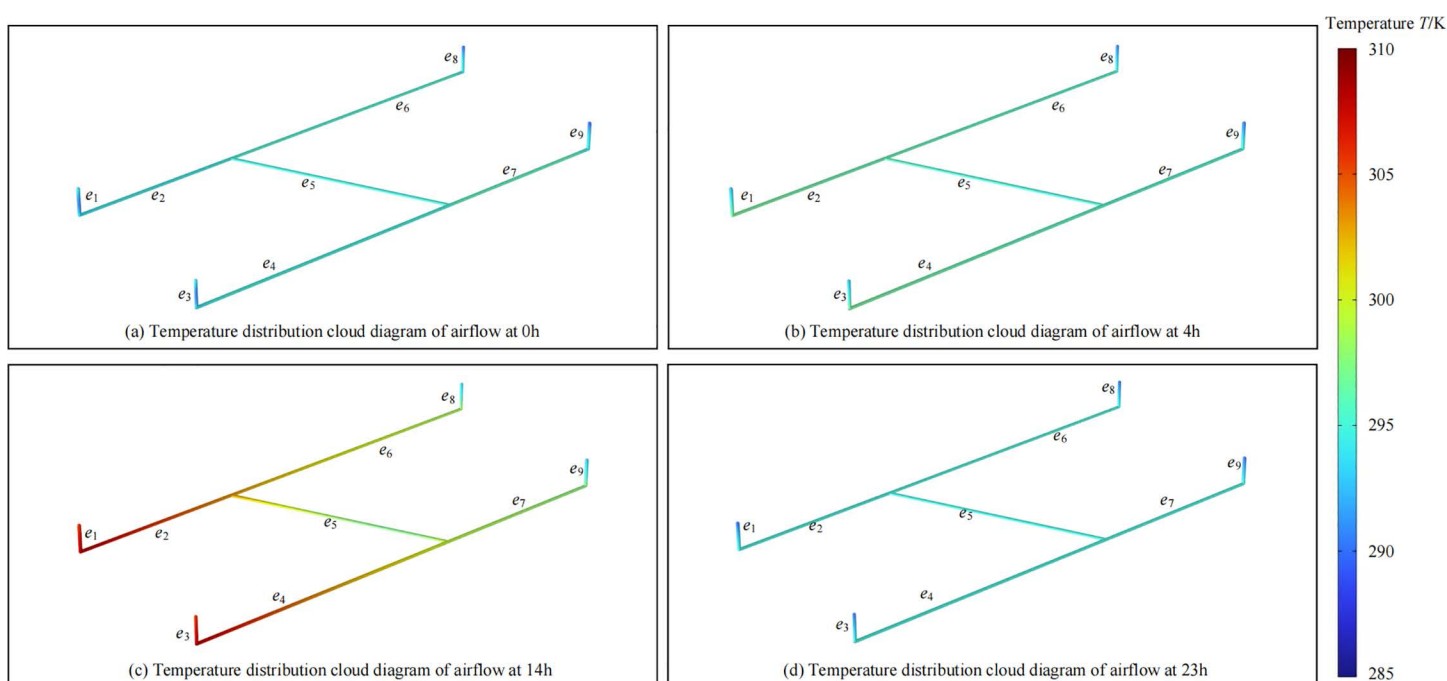

**Fig 6. Dynamic temperature distribution of airflow in mine roadways at time-series extremes.**

progresses along the shaft, the temperature difference between the airflow and the wall surface gradually decreases, reducing the intensity of heat exchange. The airflow temperature fluctuates within a smaller range and tends to stabilize. During this process, the dominant influencing factor of the airflow temperature gradually shifts to $T_2$. When the airflow enters the underground horizontal roadways $e_2$, $e_4$, $e_5$, $e_6$ and $e_7$, the temperature difference between the airflow and the roadway walls causes immediate dynamic heat exchange upon entering $e_2$ and $e_4$, significantly lowering the airflow temperature. As the airflow continues into $e_5$, $e_6$ and $e_7$, continuous heat dissipation to the roadway walls reduces the temperature difference between the airflow and the walls, gradually achieving heat equilibrium. Consequently, the airflow temperature distribution approaches the wall temperature, with the airflow temperature in the horizontal roadways primarily influenced by the underground environmental temperature field $T_2$. When the airflow enters the outlet shafts $e_8$ and $e_9$, despite the prior heat dissipation in the horizontal roadways, the airflow temperature remains higher than the outlet shaft wall temperature, causing heat transfer from the airflow to the shaft walls. This results in a gradual decrease in airflow temperature, and after flowing a certain distance, the heat exchange between the airflow and the shaft walls reaches dynamic heat equilibrium, stabilizing the airflow temperature. In the outlet airpath, the airflow temperature is mainly influenced by the underground temperature field $T_2$ and the self-compression heat of the airflow.

According to the ideal gas state equation, the airflow temperature can be considered a ternary non-monotonic function coupled with density and varying with pressure. Therefore, the dynamic changes in airflow temperature indicate that all state properties values of the airflow exhibit interlinked changes. To obtain the specific numerical distribution of the airflow state properties values and further analyze their variation patterns, an integral probe was used to collect data from the model's temperature field, pressure field, and density field. The results are listed in Table 2.

Table 2 illustrates the trend of changes in the airflow state properties values across various branches in time series. From a temperature perspective, the airflow temperature within the inlet shaft exhibits a lag compared to the atmospheric temperature changes. This lag is fundamentally due to the initially low temperature of the roadway, which causes the airflow entering the shaft at atmospheric temperature to transfer heat towards the shaft walls until heat equilibrium is reached. Subsequently, the trend of airflow temperature changes gradually aligns with the atmospheric temperature field above ground. Similarly, in horizontal roadways and the outlet shaft, the airflow temperature changes also lag behind the atmospheric temperature. However, due to non-uniform heat exchange processes underground and uneven

**Table 2. Variation data of airflow state properties in roadways in time series.**

| Branch | Presure $P$/kPa | | | | Temperature $T$/K | | | | Density $\rho$/ kg·m⁻³ | | | |
|---|---|---|---|---|---|---|---|---|---|---|---|---|
| | 0 h | 4 h | 14 h | 23 h | 0 h | 4 h | 14 h | 23 h | 0 h | 4 h | 14 h | 23 h |
| $e_1$ | 95.81 | 95.83 | 95.67 | 95.81 | 291.67 | 294.84 | 307.35 | 292.57 | 1.14 | 1.13 | 1.08 | 1.14 |
| $e_2$ | 98.13 | 98.10 | 97.81 | 98.10 | 294.36 | 296.54 | 305.65 | 294.9 | 1.16 | 1.15 | 1.12 | 1.16 |
| $e_3$ | 95.73 | 95.74 | 95.58 | 95.72 | 291.7 | 294.82 | 307.27 | 292.56 | 1.14 | 1.13 | 1.08 | 1.14 |
| $e_4$ | 98.04 | 98.00 | 97.71 | 98.00 | 294.74 | 296.24 | 303.68 | 294.92 | 1.16 | 1.15 | 1.12 | 1.16 |
| $e_5$ | 97.87 | 97.81 | 97.52 | 97.81 | 295.05 | 295.51 | 298.67 | 294.98 | 1.16 | 1.15 | 1.14 | 1.16 |
| $e_6$ | 97.47 | 97.47 | 97.18 | 97.47 | 295.27 | 295.75 | 300.32 | 294.96 | 1.15 | 1.15 | 1.13 | 1.15 |
| $e_7$ | 97.51 | 97.56 | 97.27 | 97.56 | 295.77 | 295.47 | 298.49 | 294.97 | 1.15 | 1.15 | 1.14 | 1.15 |
| $e_8$ | 93.97 | 94.07 | 93.82 | 94.07 | 292.77 | 292.93 | 295.47 | 292.5 | 1.12 | 1.12 | 1.11 | 1.12 |
| $e_9$ | 94.16 | 94.29 | 94.04 | 94.29 | 292.83 | 292.86 | 295.02 | 292.5 | 1.12 | 1.12 | 1.11 | 1.12 |

temperature distribution within internal modules, heat transfer is delayed, resulting in a generally slower trend of temperature changes across the entire domain. From a density perspective, since the humidity of the airflow and the saturated vapor pressure remain essentially constant, the gas constant can be approximated as constant. Therefore, the temperature of the airflow and its density are inversely related; as temperature increases, density decreases, with their change gradients being opposite. However, at moments when the temperature remains unchanged, the density does not show a stable trend. This is because, during periods of constant temperature, the pressure of the airflow dynamically changes under the influence of continuity. According to the combined Charles-Boyle law [22], under constant temperature conditions, density is directly proportional to pressure. Therefore, the state properties values of the airflow will exhibit highly correlated nonlinear changes under the combined influence of non-steady environmental fields and gradient flow fields. This indicates that, under the influence of non-steady environmental and flow fields, the airflow process is not an isochoric process but a polytropic process controlled by the coupling of aerodynamics and thermodynamics, essentially manifesting as heat flow characteristic.

To validate the reliability of the proposed transient heat and airflow flow characteristic model $\rho(t)$, the sum of squared residuals method was used to perform a multi-dimensional interpolation fitting analysis on the simulated pressure, temperature, and density data against $\rho(t)$. Specifically, the airflow density was taken as the characteristic analysis quantity, and the simulated pressure and temperature were used as input variables to calculate the airflow density values in time series using $\rho(t)$(i.e., the calculated data). The simulated density was used as reference data (i.e., the simulation data). By calculating the sum of squared residuals ($S_{res}$) between the calculated data and the simulation data, the total sum of squares ($S_{tot}$), and the coefficient of determination $R^2$ value, the fitting effect was evaluated to validate the reliability of the transient characteristic model $\rho(t)$. The fitting results are shown in Fig 7, and the relevant data are listed in Table 3.

The residual analysis of the interpolation fitting indicates that, within an acceptable error range, the simulated pressure, temperature, and density data conform to the transient heat flow

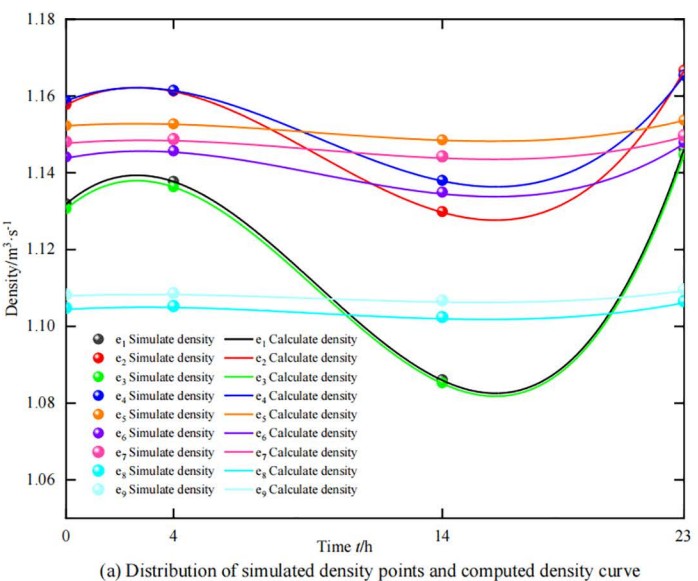
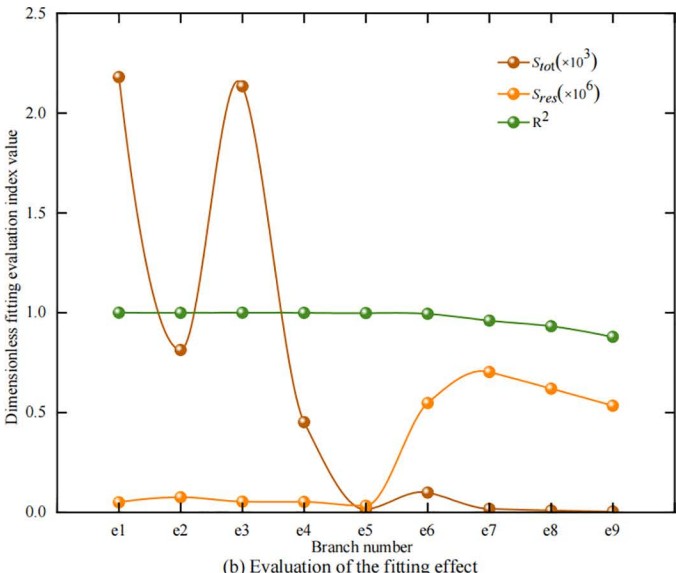

**Fig 7. Validation of transient heat and airflow flow characteristic model using multi-dimensional interpolation fitting analysis.**

**Table 3. Evaluation data for fitting effect.**

| Branch Number | Simulate density/ kg·m⁻³ | | | | Calculate density/ kg·m⁻³ | | | | Residual Sum of Squares $S_{res}$ | Total Sum of Squares $S_{tot}$ | Coefficient of determination $R^2$ |
|---|---|---|---|---|---|---|---|---|---|---|---|
| | 0 h | 4h | 14h | 23h | 0 h | 4h | 14h | 23h | | | |
| $e_1$ | 1.13 | 1.14 | 1.09 | 1.15 | 1.13 | 1.14 | 1.09 | 1.15 | $2.18 \times 10^{-3}$ | $5.00 \times 10^{-8}$ | 0.99 |
| $e_2$ | 1.16 | 1.16 | 1.13 | 1.17 | 1.16 | 1.16 | 1.13 | 1.17 | $8.14 \times 10^{-4}$ | $7.55 \times 10^{-8}$ | 0.99 |
| $e_3$ | 1.13 | 1.14 | 1.09 | 1.14 | 1.13 | 1.14 | 1.09 | 1.14 | $2.13 \times 10^{-3}$ | $5.36 \times 10^{-8}$ | 0.99 |
| $e_4$ | 1.16 | 1.16 | 1.14 | 1.17 | 1.16 | 1.16 | 1.14 | 1.17 | $4.52 \times 10^{-4}$ | $5.25 \times 10^{-8}$ | 0.99 |
| $e_5$ | 1.15 | 1.15 | 1.15 | 1.15 | 1.15 | 1.15 | 1.15 | 1.15 | $1.53 \times 10^{-5}$ | $3.30 \times 10^{-8}$ | 0.99 |
| $e_6$ | 1.14 | 1.15 | 1.13 | 1.15 | 1.14 | 1.15 | 1.13 | 1.15 | $9.94 \times 10^{-5}$ | $5.47 \times 10^{-7}$ | 0.99 |
| $e_7$ | 1.15 | 1.15 | 1.14 | 1.15 | 1.15 | 1.15 | 1.14 | 1.15 | $1.77 \times 10^{-5}$ | $7.03 \times 10^{-7}$ | 0.96 |
| $e_8$ | 1.10 | 1.10 | 1.10 | 1.11 | 1.10 | 1.11 | 1.10 | 1.11 | $9.20 \times 10^{-6}$ | $6.20 \times 10^{-7}$ | 0.93 |
| $e_9$ | 1.11 | 1.11 | 1.11 | 1.11 | 1.11 | 1.11 | 1.11 | 1.11 | $4.42 \times 10^{-6}$ | $5.34 \times 10^{-7}$ | 0.88 |

characteristics of the airflow. This result demonstrates the reliability of the proposed transient heat and airflow flow characteristic model for airflow. Furthermore, it confirms that the airflow state properties values obtained by solving this model can serve as a fundamental data source for constructing the airflow regulation model based on the heat and airflow coupling characteristics of airflow, ensuring its applicability to actual mine production environments.

## 4. Optimal solution set decision for airflow regulation in Gucheng Coal Mine

By integrating time-series data of the mine heat environment into the transient heat and airflow flow characteristic model of airflow, the fluctuating natural ventilation pressure and ventilation resistance model, as well as the airflow regulation model, the optimal airflow regulation scheme is determined. This approach aims to meet the underground ventilation requirements while minimizing the energy consumption of the system.

### 4.1. Basic ventilation parameters and heat environment information

The AGF606 axial flow fans are used in both the central and Taoyuan outlet shafts in Gucheng Coal Mine. The working blade angle of the fan in the central outlet shaft is set to 25°, and the performance curve equation is $h_a = 3255.2+10.727q_a-0.044q_a^2$. The working blade angle of the fan in the Taoyuan outlet shaft is set to 30°, and the performance curve equation is $h_b = 3372.4+9.703q_b-0.033q_b^2$.

Based on the maximum resistance routes and air supply measurements provided by Gucheng Coal Mine during the onsite survey conducted on June 25, 2023, from [10, 15]h, the resistance, airflow, and ventilation resistance of each equivalent branch at the specified times were calculated. The results are shown in Table 4. At this time, the total system power consumption of Gucheng Coal Mine was 1594.57kW

The environmental field and flow field data obtained from the actual survey during the construction of the unsteady numerical model for Gucheng Coal Mine were used as the heat environment information for the mine. However, when constructing the numerical model, considering the small variation in humidity within a single roadway, it was assumed that the phase state of water remains unchanged during the airflow process, with no latent heat of vaporization, thereby neglecting the changes in roadway humidity. To more accurately reflect the non-uniformity of the actual mine environment humidity distribution, a heat-moisture exchange heat source was introduced during the numerical calculations to simulate the heat and moisture transfer mechanisms in the underground environmental field. Since calculating

**Table 4. Data of equivalent branch ventilation network calculation.**

| Equivalent branch | Equivalent resistance $hi_{-r}$/Pa | Equivalent airflow $qi$/m³·s⁻¹ | Equivalent ventilation resistance $Ri$/N·(s²·m⁻⁸) |
|---|---|---|---|
| $e_1$ | 550.82 | 373.90 | $3.94 \times 10^{-3}$ |
| $e_2$ | 582.97 | 373.90 | $4.17 \times 10^{-3}$ |
| $e_3$ | 720.28 | 320.32 | $7.02 \times 10^{-3}$ |
| $e_4$ | 441.84 | 320.32 | $4.31 \times 10^{-3}$ |
| $e_5$ | 28.32 | 65.97 | $6.51 \times 10^{-3}$ |
| $e_6$ | 705.86 | 307.93 | $7.44 \times 10^{-3}$ |
| $e_7$ | 531.23 | 386.29 | $3.56 \times 10^{-3}$ |
| $e_8$ | 546.16 | 307.93 | $5.76 \times 10^{-3}$ |
| $e_9$ | 532.72 | 386.29 | $3.57 \times 10^{-3}$ |

the moisture exchange between the airflow and the wall involves the average wet-bulb temperature of the roadway, which is difficult to measure in production, a humidity zoning setting was adopted to represent the gradient changes in humidity. Specifically, the humidity was set as follows: 75% for the inlet shaft, 80% for the inlet roadway, 85% for the working roadway, 90% for the outlet roadway, and 95% for the outlet shaft.

## 4.2. Decision on the optimal solution set for airflow regulation

**(1) Solution of the transient heat and airflow flow characteristic of airflow in time series.** Using the unstable heat transfer coefficient of the roadway surrounding rock and the height difference of the roadway as input parameters, and comprehensively considering the heat transfer effect between the airflow and the roadway wall and the self-compression heating effect of the airflow, the hirakatsu hiramatsu's improved method [26] is employed to perform numerical iterative calculations of the airflow temperature at each node to obtain the global temperature distribution of the ventilation network.

On this basis, considering the continuous flow characteristics of the mine airflow, the pressure gradient caused by gravity effects and the pressure drop due to frictional energy losses are calculated. These are used as calculation terms, and the nodal pressure method is applied to compute the pressure distribution in the roadway system. That is, based on the airflow movement equations and the ventilation resistance loss model, a pressure iterative calculation framework is constructed. By incorporating the dynamic pressure, static pressure, and ventilation resistance effects of the airflow, the pressure values at each node are derived.

The obtained node temperatures and pressure values are substituted into the transient characteristics model of the airflow to determine the airflow density distribution at each node.

Due to the dynamic changes in the airflow state properties over time and space, to obtain the overall average state properties of the roadway airflow, numerical integration is used to integrate and sum the temperature, pressure, and density at each node. The average values are then calculated based on the spatial discretization scheme of the roadway to ensure that the results accurately reflect the heat state distribution characteristics within the roadway. The calculation results are summarized in Table 5.

**(2) Solution of the fluctuating natural ventilation pressure and ventilation resistance model in time series.** The airflow state properties values in time series are substituted into the corrected fluctuation natural ventilation pressure equation (6) and the corrected fluctuation ventilation resistance equation (8) to complete the fluctuation correction of the natural ventilation pressure and ventilation resistance for the Gucheng Coal Mine. On this basis, the corrected fluctuation values $h_N(t)'$ and $R_i'(t)$ are used as the initial values for

**Table 5. Global state properties distribution of roadways in Gucheng Coal Mine under in time series.**

| Branch | Presure $P$/kPa | | | | Temperature $T$/K | | | | Density $\rho$/kg·m⁻³ | | | |
|---|---|---|---|---|---|---|---|---|---|---|---|---|
| | 0h | 4h | 14h | 23h | 0h | 4h | 14h | 23h | 0h | 4h | 14h | 23h |
| $e_1$ | 95.92 | 95.84 | 95.83 | 95.71 | 296.46 | 294.54 | 307.87 | 292.56 | 1.11 | 1.12 | 1.06 | 1.13 |
| $e_2$ | 98.22 | 98.08 | 98.08 | 97.85 | 298.54 | 296.77 | 309.09 | 294.94 | 1.13 | 1.13 | 1.08 | 1.14 |
| $e_3$ | 95.83 | 95.75 | 95.75 | 95.63 | 296.38 | 294.49 | 307.64 | 292.54 | 1.11 | 1.12 | 1.06 | 1.13 |
| $e_4$ | 98.12 | 97.98 | 97.99 | 97.76 | 298.39 | 296.67 | 308.60 | 294.91 | 1.13 | 1.13 | 1.08 | 1.14 |
| $e_5$ | 97.92 | 97.78 | 97.79 | 97.56 | 298.20 | 296.59 | 307.75 | 294.94 | 1.13 | 1.14 | 1.09 | 1.14 |
| $e_6$ | 97.58 | 97.46 | 97.47 | 97.25 | 298.21 | 296.60 | 307.81 | 294.94 | 1.13 | 1.14 | 1.08 | 1.14 |
| $e_7$ | 97.63 | 97.51 | 97.52 | 97.30 | 298.14 | 296.54 | 307.60 | 294.91 | 1.13 | 1.14 | 1.09 | 1.14 |
| $e_8$ | 94.15 | 94.07 | 94.06 | 93.98 | 295.44 | 294.00 | 304.00 | 292.52 | 1.11 | 1.12 | 1.07 | 1.12 |
| $e_9$ | 94.30 | 94.21 | 94.20 | 94.12 | 295.47 | 294.00 | 304.22 | 292.48 | 1.11 | 1.12 | 1.07 | 1.13 |

iteration. Using continuous time as the allocation frequency, the cross algorithm is employed to perform reverse iteration to calculate the airflow volumes $q_i(t)'$ in each branch of the ventilation network in time series. The solution of the fluctuating natural ventilation pressure and ventilation resistance model in time series is thus completed, and the results are listed in Tables 6 and 7.

(3) **Solution of the airflow regulation model in time series.** the nodes $V_1$ and $V_3$ are exposed to the same atmospheric temperature conditions, they can be merged into a virtual node $V_m$. To enhance the connectivity of the ventilation network, virtual branches $e_{10}$ and $e_{11}$ are introduced, leading from the fan outlet to $V_m$. Consequently, the ventilation network of Gucheng Coal Mine can be made equivalent to the one depicted in Fig 8. The air volume in these virtual branches equals the fan air volume, and the resistance is governed by the fan performance curve equation.

The diagram shows a total of three airpaths in the Gucheng Coal Mine. The fan-controlled airpath for the central outlet shaft $e_8$ is as follows: $e_1 \to e_2 \to e_6 \to e_8$. The fan-controlled airpath for the Taoyuan outlet shaft $e_9$ comprises two segments: $e_1 \to e_2 \to e_5 \to e_7 \to e_9$ and $e_3 \to e_4 \to e_7 \to e_9$. There are a total of 4 closed loops (including virtual branches):

$$c_1 = \{e_1, e_2, -e_3, -e_4, e_5\},$$

**Table 6. Time series data of fluctuating natural ventilation pressure and frictional resistance.**

| Branch | The corrected Ri(t)'/N·s2·m-8 | | | | Fan | The correct natural ventilation pressure $h_N(t)'$/Pa | | | |
|---|---|---|---|---|---|---|---|---|---|
| | 0h | 4h | 14h | 23h | | 0h | 4h | 14h | 23h |
| $e_1$ | 3.81×10⁻³ | 3.84×10⁻³ | 3.65×10⁻³ | 3.87×10⁻³ | central | -147.17 | -138.42 | -171.28 | -127.63 |
| $e_2$ | 4.04×10⁻³ | 4.06×10⁻³ | 3.86×10⁻³ | 4.09×10⁻³ | Taoyuan | -142.26 | -133.67 | -163.51 | -123.25 |
| $e_3$ | 6.80×10⁻³ | 6.85×10⁻³ | 6.51×10⁻³ | 6.89×10⁻³ | | | | | |
| $e_4$ | 4.18×10⁻³ | 4.20×10⁻³ | 4.00×10⁻³ | 4.23×10⁻³ | | | | | |
| $e_5$ | 6.38×10⁻³ | 6.41×10⁻³ | 6.11×10⁻³ | 6.44×10⁻³ | | | | | |
| $e_6$ | 7.31×10⁻³ | 7.35×10⁻³ | 7.01×10⁻³ | 7.39×10⁻³ | | | | | |
| $e_7$ | 3.50×10⁻³ | 3.52×10⁻³ | 3.36×10⁻³ | 3.54×10⁻³ | | | | | |
| $e_8$ | 5.72×10⁻³ | 5.75×10⁻³ | 5.51×10⁻³ | 5.78×10⁻³ | | | | | |
| $e_9$ | 3.55×10⁻³ | 3.56×10⁻³ | 3.41×10⁻³ | 3.58×10⁻³ | | | | | |

**Table 7. Time series data of fluctuating global air volume and ventilation resistance.**

| Branch | Iterative correction air volume $q(t)'/m^3 \cdot s^{-1}$ | | | | Iterative correction resistance $h_{i\text{-}r}(t)'/Pa$ | | | |
|---|---|---|---|---|---|---|---|---|
| | 0h | 4h | 14h | 23h | 0h | 4h | 14h | 23h |
| $e_1$ | 370.40 | 370.17 | 373.12 | 370.22 | 523.38 | 526.19 | 508.56 | 529.81 |
| $e_2$ | 370.40 | 370.17 | 373.12 | 370.22 | 554.25 | 556.87 | 537.32 | 560.10 |
| $e_3$ | 317.24 | 317.02 | 319.41 | 317.06 | 684.46 | 688.03 | 664.64 | 692.72 |
| $e_4$ | 317.24 | 317.02 | 319.41 | 317.06 | 420.59 | 422.49 | 407.80 | 424.88 |
| $e_5$ | 65.56 | 65.45 | 65.91 | 65.57 | 27.42 | 27.46 | 26.56 | 27.69 |
| $e_6$ | 304.84 | 304.72 | 307.21 | 304.65 | 679.63 | 682.72 | 661.28 | 685.54 |
| $e_7$ | 382.80 | 382.47 | 385.32 | 382.63 | 513.27 | 515.06 | 498.79 | 517.80 |
| $e_8$ | 304.84 | 304.72 | 307.21 | 304.65 | 531.91 | 534.17 | 519.64 | 536.41 |
| $e_9$ | 382.80 | 382.47 | 385.32 | 382.63 | 519.71 | 521.43 | 506.50 | 524.32 |

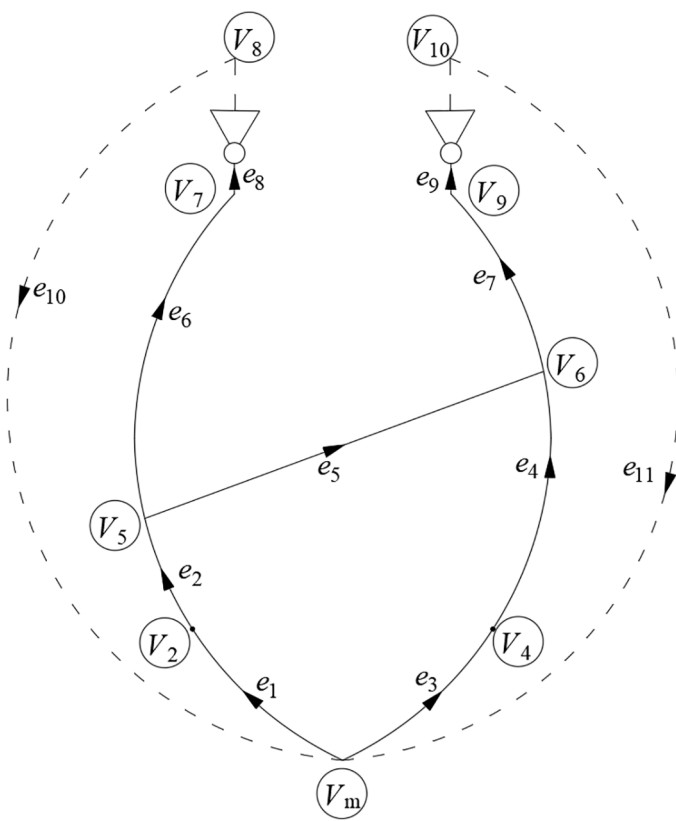

**Fig 8. Equivalent diagram of ventilation network in Gucheng Coal Mine.**

$$c_2 = \{e_1, e_2, e_6, e_8, e_{10}\},$$

$$c_3 = \{e_3, e_4, e_7, e_9, e_{11}\},$$

$$c_4 = \{e_1, e_2, e_5, e_7, e_9, e_{11}\},$$

Based on the distribution of branches in the closed loops, the fundamental loop matrix for the Gucheng Coal Mine is constructed. By taking the branch adjustment resistance as the decision variable $x$, and integrating the airflow data $q_i(t)'$ from each branch in the ventilation network in time series, a coefficient matrix $\mathbf{A}_{eq}$, corresponding to different time steps is formed. Simultaneously, the unbalanced ventilation pressure values of the loops at each time step are used as the constraint constant vector $\mathbf{b}_{eq}$, to complete the construction of the standard linear programming model. Since $q_i(t)'$ is part of the coefficient matrix involved in the overall optimization process and exhibits a high coupling relationship with temperature changes, with temperature extremum points corresponding to sensitive moments of airflow fluctuations, the time series extremum points [0, 4, 14, 23]h are chosen as the model calculation time steps. The simplex method is used to solve the linear programming model, determining the optimal branch adjustment scheme for each time step. The specific airflow regulation scheme is as follows.

**Scheme 1**

$$\left(t \,=\, 0h\right): \Delta h_3 = 7.89 \times 10^{-4} P, \Delta h_5 = 6.47 \times 10^{-3} Pa, \Delta h_6 = 2.38 \times 10^{-3} Pa$$

$$\Delta R_{3\_} \Delta h_3 / q_3(0)'^2 = 7.89 \times 10^{-4} / 317.24^2 = 7.84 \times 10^{-9}$$

$$\Delta R_{5\_} \Delta h_5 / q_5(0)'^2 = 6.47 \times 10^{-3} / 65.56^2 = 1.51 \times 10^{-6}$$

$$\Delta R_{6\_} \Delta h_6 / q_6(0)''^2 = 2.38 \times 10^{-3} / 304.84^2 = 2.56 \times 10^{-8}$$

Based on the calculation results, the branch friction resistance vector is updated to complete the second calculation of the ventilation network. Table 8 presents the ventilation network calculation result.

At this time,

$$minN_1 = h_{N\_Central}(0)' \Delta q_8(0)'' + h_{N\_Taoyuan}(0)' \Delta q_9(0)'' + \sum h_{i-r}(0)'' q_i(0)'' = 1416.96 kW;$$

$$minN_2 = \Delta h_3 \Delta q_3(0)'' + \Delta h_5 \Delta q_5(0)'' + \Delta h_6 \Delta q_6(0)'' = 1.40 \times 10^{-3} kW;$$

$$minN = minN_1 + minN_2 = 1416.96 kW$$

**Scheme 2**

$$\left(t = 4h\right): \Delta h_3 = 4.76 Pa, \Delta h_5 = 4.76 Pa, \Delta h_8 = -1.24 \times 10^{-2} Pa$$

$$\Delta R_{3\_} \Delta h_3 / q_3(4)'^2 = 4.76 / 317.02^2 = 4.74 \times 10^{-5}$$

$$\Delta R_{5\_} \Delta h_5 / q_5(4)'^2 = 4.76 \times 10^{-3} / 65.45^2 = 1.11 \times 10^{-3}$$

$$\Delta R_{8\_} \Delta h_8 / q_8(4)'^2 = -1.24 \times 10^{-2} / 304.72^2 = -1.33 \times 10^{-7}$$

Based on the calculation results, the branch friction resistance vector is updated to complete the second calculation of the ventilation network. Table 8 presents the ventilation network calculation result.

At this time,

$$minN = minN_1 + minN_2 = 1429.40 + 1.82 = 1431.22 kW$$

**Scheme 3**

$$(t = 14h): \Delta h_1 = -4.17 \times 10^{-3} Pa, \Delta h_3 = -4.70 \times 10^{-3} Pa, \Delta h_5 = 3.23 \times 10^{-3} Pa$$

$$\Delta R_{1=} \Delta h_1 / q_1 (14)'^2 = -2.99 \times 10^{-8}$$

$$\Delta R_{3=} \Delta h_3 / q_3 (14)'^2 = -4.61 \times 10^{-8}$$

$$\Delta R_{5=} \Delta h_5 / q_5 (14)'^2 = 7.44 \times 10^{-7}$$

Based on the calculation results, the branch friction resistance vector is updated to complete the second calculation of the ventilation network. Table 8 presents the ventilation network calculation result.

At this time,

$$minN = minN_1 + minN_2 = 1369.06 - 2.84 \times 10^{-3} = 1369.06 kW.$$

**Scheme 4**

$$(t = 23h): \Delta h_1 = -1.26 \times 10^{-2} Pa, \Delta h_4 = -2.18 \times 10^{-3} Pa, \Delta h_5 = 1.03 \times 10^{-2} Pa$$

$$\Delta R_{1=} \Delta h_1 / q_1 (23)'^2 = -9.16 \times 10^{-8}$$

**Table 8. Ventilation network calculation data under different resistance regulation schemes.**

| Branch $j$ | Scheme 1 | | Scheme 2 | | Scheme 3 | | Scheme 4 | |
|---|---|---|---|---|---|---|---|---|
| | Update air volume $q(t)''/\text{m}^3\cdot\text{s}^{-1}$ | Update ventilation resistance $h_{i\text{-}r}(t)''/\text{Pa}$ | Update air volume $q(t)''/\text{m}^3\cdot\text{s}^{-1}$ | Update ventilation resistance $h_{i\text{-}r}(t)''/\text{Pa}$ | Updateair volume $q(t)''/\text{m}^3\cdot\text{s}^{-1}$ | Update ventilation resistance $h_{i\text{-}r}(t)''/\text{Pa}$ | Update air volume $q(t)''/\text{m}^3\cdot\text{s}^{-1}$ | Update ventilation resistance $h_{i\text{-}r}(t)''/\text{Pa}$ |
| $e_1$ | 370.40 | 523.38 | 370.08 | 525.94 | 373.12 | 508.56 | 370.22 | 529.80 |
| $e_2$ | 370.40 | 554.25 | 370.08 | 556.61 | 373.12 | 537.32 | 370.22 | 560.10 |
| $e_3$ | 317.24 | 684.46 | 316.93 | 692.40 | 319.41 | 664.64 | 317.06 | 692.72 |
| $e_4$ | 317.24 | 420.59 | 316.93 | 422.26 | 319.41 | 407.80 | 317.06 | 424.88 |
| $e_5$ | 65.56 | 27.42 | 65.34 | 32.11 | 65.91 | 26.56 | 65.57 | 27.70 |
| $e_6$ | 304.84 | 679.64 | 304.74 | 682.81 | 307.21 | 661.28 | 304.65 | 685.54 |
| $e_7$ | 382.80 | 513.27 | 382.27 | 514.53 | 385.32 | 498.79 | 382.63 | 517.80 |
| $e_8$ | 304.84 | 531.91 | 304.74 | 534.23 | 307.21 | 519.64 | 304.65 | 536.41 |
| $e_9$ | 382.80 | 519.71 | 382.27 | 520.89 | 385.32 | 506.50 | 382.63 | 524.32 |

$$\Delta R_{4=}\Delta h_4 / q_4 (23)'^2 = -2.17 \times 10^{-8}$$

$$\Delta R_{5=}\Delta h_5 / q_5 (23)'^2 = 2.40 \times 10^{-6}$$

Based on the calculation results, the branch friction resistance vector is updated to complete the second calculation of the ventilation network. Table 8 presents the ventilation network calculation result.

At this time,

$$minN = minN_1 + minN_2 = 1444.63 - 4.66 \times 10^{-3} = 1444.62 kW$$

By solving the airflow regulation model in time series, the optimal branch adjustment schemes for each time step were determined. According to the system power consumption calculation results, it is evident that within the time intervals [0, 4]h and [14, 23]h, the temperature gradually decreases, and the air volume also decreases. However, the system's power consumption shows an increasing trend. Conversely, within the time interval [4, 14]h, the temperature gradually increases, but the system's power consumption decreases, reaching its lowest value when the temperature reaches its maximum point ($t$=14h). This further reveals the nonlinear coupling relationship between the state properties values of the airflow and the ventilation parameters. Although the temperature decreases and the air volume reduces, the increase in ventilation resistance and the decrease in natural ventilation pressure require the system to consume more energy to maintain ventilation stability. Therefore, the interaction between the state properties values of the airflow and the ventilation parameters is influenced by multiple factors rather than a single one, with the fluctuations in natural ventilation pressure and ventilation resistance intensifying this coupling effect, resulting in complex variations in power consumption over different time periods.

By optimizing the branch adjustment schemes in the ventilation network, it is possible to effectively reduce the overall system power consumption while ensuring the airflow requirements of each branch, thereby achieving more economical and efficient ventilation operations. This provides data support and theoretical basis for further energy efficiency improvements and system regulation.

## 5. Conclusions

1) To precise airflow control, three models are sequentially proposed: a transient heat and airflow flow characteristics model of airflow to quantitatively describe airflow state variations in time series; a fluctuating natural ventilation pressure and ventilation resistance model to quantitatively characterize their fluctuation patterns and impacts on ventilation systems; and an airflow regulation model to quantitatively determine optimal airflow regulation solution in time series.

2) The constructed time-series-based airflow regulation model incorporating heat and airflow coupling characteristic can accurately capture dynamic changes in airflow state properties and ventilation parameters. By real-time quantification of fluctuations in natural ventilation pressure and ventilation resistance caused by dynamic mine environmental characteristics, the model enables optimal real-time airflow regulation decisions at different time steps, achieving precise real-time ventilation network airflow regulation. This real-time precise airflow regulation reduces unnecessary air volume waste, thereby decreasing ventilation system power consumption and enhancing overall system economic efficiency.

3) Real-time precise airflow regulation dynamically responds to mine environmental changes, effectively improving the adaptive control accuracy and response speed of ventilation systems while optimizing control energy efficiency of the ventilation network. This approach provides crucial data support for the automation and intelligent development of mine intelligent ventilation systems, establishing a technical foundation for intelligent mine ventilation through dynamic regulation mechanisms and time-series-based optimization methods.

## Author contributions

**Conceptualization:** Tong Jia, Ma Heng.

**Data curation:** Tong Jia, Ma Heng.

**Formal analysis:** Tong Jia.

**Investigation:** Tong Jia.

**Methodology:** Tong Jia, Ma Heng, Ke Gao.

**Resources:** Ma Heng, Ke Gao.

**Supervision:** Ma Heng, Ke Gao.

**Validation:** Tong Jia, Ma Heng.

**Writing – original draft:** Tong Jia.

**Writing – review & editing:** Ke Gao.

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
