## [Decision Letter · Decision Letter 0]

17 Dec 2024

PONE-D-24-50720Correction of natural ventilation pressure in mines based on heat–flow coupling characteristics of underground airflowPLOS ONE

Dear Dr. Heng,

Thank you for submitting your manuscript to PLOS ONE. After careful consideration, we feel that it has merit but does not fully meet PLOS ONE’s publication criteria as it currently stands. Therefore, we invite you to submit a revised version of the manuscript that addresses the points raised during the review process.

We look forward to receiving your revised manuscript.

Kind regards,

Rajeev Singh

Academic Editor

PLOS ONE

3. Thank you for stating the following in the Competing Interests section: [We declare no financial or personal relationships with other people or organisations that could unduly influence our work. We have no professional or other personal interest of any nature or kind in any product, service, and/or company that could be construed as influencing the position presented in, or the review of, the manuscript titled, ‘Correction of natural ventilation pressure in mines based on heat–flow coupling characteristics of underground airflow’.]. Please confirm that this does not alter your adherence to all PLOS ONE policies on sharing data and materials, by including the following statement: "This does not alter our adherence to  PLOS ONE policies on sharing data and materials.” (as detailed online in our guide for authors http://journals.plos.org/plosone/s/competing-interests).  If there are restrictions on sharing of data and/or materials, please state these. Please note that we cannot proceed with consideration of your article until this information has been declared.

4. We note that your Data Availability Statement is currently as follows: [All relevant data are within the manuscript and its Supporting Information files.] Please confirm at this time whether or not your submission contains all raw data required to replicate the results of your study. Authors must share the “minimal data set” for their submission. PLOS defines the minimal data set to consist of the data required to replicate all study findings reported in the article, as well as related metadata and methods (https://journals.plos.org/plosone/s/data-availability#loc-minimal-data-set-definition).

Additional Editor Comments (if provided):

Reviewers' comments:

Reviewer's Responses to Questions

**Comments to the Author**

1. Is the manuscript technically sound, and do the data support the conclusions?

Reviewer #1: Yes

Reviewer #2: Yes

Reviewer #3: Partly

2. Has the statistical analysis been performed appropriately and rigorously? 

Reviewer #1: Yes

Reviewer #2: Yes

Reviewer #3: N/A

3. Have the authors made all data underlying the findings in their manuscript fully available?

Reviewer #1: No

Reviewer #2: Yes

Reviewer #3: Yes

4. Is the manuscript presented in an intelligible fashion and written in standard English?

Reviewer #1: Yes

Reviewer #2: Yes

Reviewer #3: No

5. Review Comments to the Author

Reviewer #1: Reviewer comments

The manuscript addresses an important issue related to deep mining ventilation, particularly the mismatch between the ventilation power parameters and the underground ventilation network due to neglecting unsteady heat transfer between the airflow and the thermal environment. This mismatch makes it difficult to maintain the balance of airflow supply and demand in the mine roadways. The authors, based on the unsteady-state heat transfer leading to the fundamental nature of heat flow in airflow, propose a thermally corrected natural ventilation pressure formula and use this corrected pressure as an environmental variable to build a mathematical model for airflow control. By applying this model to the Gucheng coal mine, management decisions regarding airflow control in terms of ventilation power and resistance are made based on the model results. The research contributes to the optimization of intelligent ventilation and supplements the basic theory of the ventilation network under the heat-flow coupling characteristic of airflow, which is a novel approach. However, there are several parts in the manuscript that need to be addressed:

1. In Section 2.2, a numerical model of unsteady heat transfer for airflow in the Gucheng Coal Mine is presented. It is recommended to incorporate field measurement data or experimental data to validate the accuracy of the simulation model.

2. In Section 2.3, The proposed airflow control model, which is based on the heat-flow coupling characteristic of airflow, should be compared with currently commonly used control models. It is suggested that the proposed model be compared with commonly used control models to clearly explain the advantages and disadvantages of the new model.

3. In Section 3.1, there are errors in the notation of Equations 14 and 15; please review and correct them. In Section 3.2, the ventilation resistance calculation for the central ventilation network and Taoyuan ventilation network in Scheme 2.1 are incorrectly written; please check and amend accordingly.

4. Figures 8 and 10 represent the fan characteristic curves. The working conditions and blade angles should be clearly labeled in the figures to help readers observe the fan's status before and after adjustment.

5. The units throughout the paper should be consistent, and variables in the equations should be defined. The figures and textual descriptions should be as clear and understandable as possible. Some figures are unclear; it is advised that the author revise them to improve clarity.

6. In the conclusion section, while the main findings and contributions of the study should be clearly stated, the practical significance of these findings for the design, operation, and decision-making of intelligent ventilation systems should also be highlighted. The paper currently lacks a detailed discussion on the practical implications of the work, which should be addressed in the conclusion.

I hope the authors consider these points to improve the clarity and overall quality of the paper.

Reviewer #2: Reviewer comments

To address the issue of inadequate matching between the mine ventilation operating condition and airflow demand in the control network, which arises from the heat-flow coupling characteristic of airflow, the authors propose that the primary factors causing the mismatch between the fan operating condition and underground airflow demand are the natural ventilation pressure and mine ventilation resistance, calculated solely based on aerodynamic forces. On this basis, they developed a mathematical model for airflow regulation that incorporates the heat-flow coupling characteristics of airflow under the combined effects of aerodynamic and thermodynamic forces. The model was solved using the particle swarm optimization algorithm and the simplex method. Based on the solution results, the authors propose adjustment strategies from both ventilation power and resistance perspectives to improve the compatibility between fan operating conditions and underground airflow demand. This research has significant potential for practical applications. However, before the manuscript can be formally accepted, some minor issues need to be addressed:

1) In Section 3.1, the authors perform transient solving of the non-steady-state heat transfer model for the coal mine. How were the continuous data for temperature, pressure, and time in Table 2 derived?

2) The grid has a significant impact on the results. In the grid independence verification for the heat transfer model, the grid evaluation parameters used in Figure 6 continue to rise as the number of grids increases. An additional analysis of the selected grid division scheme should be included. It is also suggested to adjust the content of Figure 6 to ensure that the curve remains complete.

3) How is the proposed airflow regulation model applied in conjunction with CFD numerical simulation software? Please provide more details.

4) The authors should consider the potential discrepancies between the simulation results and actual conditions, and evaluate the effectiveness of the simulation. More details should be provided, including the methods of data collection, as well as a thorough analysis of the experimental results, so that readers can assess the reliability and validity of the study.

5) The models and data results mentioned in the article should be presented clearly through figures and tables to aid reader comprehension. Each figure and table should include clear titles, legends, and annotations to facilitate interpretation.

6) The formatting of the article, such as line spacing and other elements, should be adjusted to ensure consistency throughout the manuscript. It is recommended to carefully review the entire document, including text, figures, and images, to ensure uniform formatting.

7) The conclusions do not fully summarize the research content and its innovations. It is suggested to refine and strengthen the conclusions to more effectively highlight the key contributions and novel aspects of the study.

Reviewer #3: The manuscript explores the density-pressure coupling rule's impact on airflow in a mechanically ventilated mine. The issue is interesting and should be published. However, the problem of the influence of natural ventilation pressure (NVP) on the airflow in a mine ventilation network and the characteristics of the fan's operation has been known and described in the literature since at least the first half of the last century.

Authors are reminded that in the introduction, they should refer to articles describing the influence of the compressibility of air NVP in mine ventilation networks. Many publications have been written on this subject. Many methods describe the determination of NVP in a mine ventilation network. Additionally, the work analyzes the influence of NVP on the operation of the main ventilation fan. Therefore, the article's title should indicate that this is a different way of approaching this issue. The title could sound similar to "Heat-airflow-coupled approach to correcting the influence of the natural ventilation pressure on the airflow in an underground mine's ventilation network."

Other major comments:

1) To avoid any confusion, it is recommended that the term' heat and airflow coupling' be used to clearly indicate the relationship between heat and airflow, rather than being associated with heat flow only.

2) The authors should remember that the resistance of a single branch and the total resistance of the ventilation network are functions of the variable air density. Therefore, it is recommended that the standardized resistance of branches and the standardized fan characteristics be considered. This understanding is crucial as it significantly impacts the results of the analysis at the Gucheng Coal Mine.

3) The particle swarm optimization algorithm for solving the ventilation network should be referred to the reference or described in more detail.

4) The term "frictional air resistance" should be changed to the total resistance of branches, equivalent to the resistance of branches in the real mine.

5) Mine ventilation power is a strictly defined term in the science of mine ventilation. The authors probably use this term for a different purpose. Ventilation power and the electric power of engines are different parameters. It is proposed to define the term "mine ventilation" in the article.

6) Similarly, "system power consumption" and "the flow power of the aerodynamic force" should be defined by appropriate formulas so that the reader can easily understand the article.

7) It is proposed that chapter 8 in McPherson's book in this manual be cited in the references.

8) In conclusion 1, the number of the developed formula should be given.

9) The first sentence of Conclusion 2 needs to be simplified and should be rewritten more clearly and comprehensively to ensure the reader's full understanding.

10) Conclusion 3 has not been proven to maintain a stable flow condition, and flow stability has yet to be defined.

Other minor comments

1) The term "stable airflow conditions: is used in the wrong context of the text.

2) What does "state equation" mean when giving an approximation equation of the fan characteristic curve?

3) The caption of Figure 7 needs to be corrected. The temperature color range is too extensive.

4) It is recommended that the symbols used in the manuscript be standardized.

5) English needs proofing.

It is recommended to study the publication (DOI):

https://hdl.handle.net/10520/AJA0038223X_5533 10.2478/amsc-2014-0036

10.1007/BF01560715

10.1134/S1062739147050145

10.19835/j.issn.1008-4495.2023.04.017

10.1016/j.ijmst.2017.09.004

10.1155/2022/8789955

10.1080/15567036.2019.1673512

6. PLOS authors have the option to publish the peer review history of their article (what does this mean? ). If published, this will include your full peer review and any attached files.

**Do you want your identity to be public for this peer review?** For information about this choice, including consent withdrawal, please see our Privacy Policy .

Reviewer #1: No

Reviewer #2: No

Reviewer #3: No

---

## [Author Response · Author response to Decision Letter 1]

11 Feb 2025

Reviewer Comments & Revision Explanation

Dear Editors and Reviewers,

Thank you for your thoughtful review of my manuscript. I apologize for the delay in submitting this revised version. During the revision process, I have conducted a detailed collection and secondary utilization of data, particularly focusing on the numerical simulations and practical application calculations that were highlighted in the reviewer comments. This process took longer than anticipated. I kindly request that you review the revised manuscript, and I look forward to your valuable feedback.

I. Revisions Regarding Journal Requirements

Thank you for your diligent work and thorough review of my manuscript. This is my first submission to PLOS ONE, and I acknowledge that there were several aspects of my manuscript that did not fully comply with the journal’s requirements. I have carefully reviewed the journal guidelines and have made revisions to both the formatting and content of the manuscript to meet the necessary standards. The revised version of the manuscript, with all adjustments, is included in the resubmitted files for your review.

I greatly appreciate the time and effort you have dedicated to reviewing my work, and I look forward to your further feedback.

Q1: Ensure that the manuscript meets PLOS ONE's style requirements.

A1: I have carefully reviewed the PLOS ONE style guidelines and ensured that the manuscript meets all formatting requirements. The manuscript has been reformatted according to the templates provided in the links:

Main Body Formatting: The manuscript has been adjusted to align with the style requirements outlined in the sample for the main body.

Title and Author Affiliations: The title page and author affiliations have been formatted according to the provided template.

I have attached the revised manuscript with the appropriate formatting.

Q2: Ensure that the code is shared in a way that follows best practice and facilitates reproducibility and reuse.

A2: The methods described in the manuscript utilize Matlab's built-in functions and toolboxes for model solving and optimization, and no custom, author-generated code was developed for this study. As a result, there are no specific code contributions that require sharing under PLOS ONE's guidelines.

However, if further clarification is needed or if any other materials are required to ensure reproducibility, I am happy to provide additional information upon request.

Q3: Please include your updated Competing Interests statement in your cover letter.

A3: I confirm that the statement in the Competing Interests section does not alter our adherence to all PLOS ONE policies on sharing data and materials. We fully commit to sharing the data and materials in accordance with PLOS ONE's policies. If there are any restrictions on sharing, they will be clearly stated in the manuscript.

The updated Competing Interests statement is as follows:

“The authors declare that they have no competing interests exist.”

I have included this updated statement in the cover letter, as requested.

Q4: Confirm at this time whether or not your submission contains all raw data required to replicate the results of your study.

A4: I confirm that the manuscript includes all relevant data to replicate the results of the study.

I confirm that all relevant data required to replicate the results of the study are included in the manuscript. Specifically, the manuscript provides all necessary data such as the values behind the means, standard deviations, and other measures, as well as the data used to construct the graphs and any points extracted from images for analysis.

If you require any additional information or further clarification on the data, please do not hesitate to let me know.

Q5: ORCID iD Requirement

A5: I have obtained an ORCID iD and have successfully validated it in Editorial Manager. Please let me know if there are any further steps I need to take regarding this.

Ⅱ. Revisions Regarding Reviewer comments

Thank you for the time and effort you dedicated to reviewing my manuscript. Your valuable suggestions have provided me with new insights on how to improve the article and guide my future research. In my response, I will address each of your comments individually, and you can find my revisions and explanatory notes on the structural corrections in the resubmitted document.

The core issue of the research is the real-time and precise regulation of airflow volume in the ventilation network. The main focus is that the traditional airflow regulation model, which considers natural ventilation pressure and ventilation resistance as constant values, making it unable to determine the optimal solution set for airflow regulation. Consequently, precise airflow regulation cannot be achieved. The key to precise ventilation control is solving the fluctuating airflow state properties in time series based on the heat and airflow coupling characteristics, deriving the fluctuating natural ventilation pressure and ventilation resistance, and applying a nonlinear correction to the traditional airflow regulation model to obtain the optimal solution set.

The main research focus of Chapter 2 is to propose a density-pressure-temperature coupling equation that describes the variation patterns of airflow state properties, forming a transient heat and airflow flow characteristics model in time series. By incorporating this characteristics model as a correction variable, further develops a fluctuating natural ventilation pressure and ventilation resistance model in time series. Finally, based on the heat and airflow coupling characteristics of airflow, a nonlinear airflow regulation model is constructed to optimize airflow regulation.

Chapter 3 focuses on verifying the reliability of the proposed transient heat and airflow flow characteristics model and the heat and airflow coupling characteristics of airflow. This is achieved through CFD simulation, solving a full-scale unsteady-state heat transfer model based on the complex environmental and gradient flow fields of an actual mine.

Chapter 4 applies the research findings to a real-world case study, using the Gucheng Coal Mine as the research subject. Field-surveyed complex environmental and gradient flow fields are used as constraints and are integrated into the proposed transient heat and airflow flow characteristics model and fluctuating natural ventilation pressure and ventilation resistance model. This allows for the calculation of airflow state properties values and fluctuating values of natural ventilation pressure and ventilation resistance. The computed values are then input into the airflow regulation model to determine the optimal ventilation regulation scheme for the Gucheng Coal Mine in time series. This enables real-time and precise control of airflow volume in the ventilation network, thereby validating the core issue of the research.

ⅡA). Reviewer #1

Q1: In Section 2.2, a numerical model of unsteady heat transfer for airflow in the Gucheng Coal Mine is presented. It is recommended to incorporate field measurement data or experimental data to validate the accuracy of the simulation model.

A1: I conducted field measurements and calculations of the length, area, perimeter, height difference, roadways temperature, and frictional resistance of underground roadways before performing simulations. I also monitored and calculated the atmospheric pressure, atmospheric temperature, and the air pressure and air volume of the fans. Based on these data, I constructed the simulation model, which can be considered accurate within an acceptable error range. However, I did not mention the model parameter values in the article. In response to your suggestion, I have compiled the data and added a basic parameter table in Section 3.2 of the article (as shown below).

Table 1. Initial physical conditions of model

Equivalent branch Function Length

L /m Altitude difference

Z /m Equivalent area

S /m2 Equivalent hydraulic diameter

dh /m Thermal conductivity of wall layer

λk /(W·m-1·K-1) Surface roughness

η /m

e1 the central inlet shaft 500 500 34.11 6.59 2.75 7.02×10-1

e2 the inlet airpath for the eastern and western areas of the mine from the central inlet shaft 3000 0 32.37 6.42 0.31 2.59×10-3

e3 the Taoyuan inlet shaft 500 500 29.51 6.13 2.75 9.40×10-1

e4 the Taoyuan inlet shaft to the southern inlet airpath 5000 0 31.77 6.36 0.31 1.33×10-4

e5 the central inlet shaft to the southern inlet airpath 3910 0 28.18 5.99 0.31 6.40×10-4

e6 the outlet airpath from the eastern and western areas to the central outlet shaft 5000 0 29.42 4.81 0.31 1.07×10-3

e7 the southern outlet airpath 3000 0 33.08 5.10 0.31 1.66×10-3

e8 the central outlet shaft 500 -500 64 8 2.75 1.03

e9 the Taoyuan outlet shaft 500 -500 64 8 2.75 9.44×10-1

During the actual survey, it was found that most of the main roadways underground were coal roadways. Therefore, when modeling, the wall boundary material of the shaft was set as siltstone, and the boundary material of the other roadways was set as lean coal.

The parameter differences between coal roadways and rock roadways are reflected in the heat transfer parameters and wall layer temperature of the roadway. The most direct heat transfer parameter affecting the convective heat transfer between the airflow and the roadway is the thermal conductivity of the wall layer, λk , and the original temperature of the wall layer (since the unstable heat transfer coefficient needs to be calculated through λk . The unstable heat transfer coefficient is the main calculation item for calculating the heat exchange magnitude between the airflow and the wall layer. Other parameters such as specific heat and density are automatically coupled and set when the material is modeled, and do not need to be assigned separately).

I have listed the thermal conductivity parameters, λk, of the wall layer in the table. It can be seen from the table that there is a significant difference in the thermal conductivity between the coal layer and the rock layer. In Section 3.3: Boundary Condition Settings, I have drawn the distribution of the underground temperature field. From the figure, it can be seen that under the influence of geothermal heat flow, the temperature setting of the coal roadway is higher than that of the rock roadway.

Fig.5 Temperature distribution in mine roadways under the underground environmental field

Q2: In Section 2.3, The proposed airflow control model, which is based on the heat-flow coupling characteristic of airflow, should be compared with currently commonly used control models. It is suggested that the proposed model be compared with commonly used control models to clearly explain the advantages and disadvantages of the new model.

A2: In traditional ventilation system analysis, airflow is treated as an incompressible turbulent fluid, and the airflow in roadways is studied as independent entities. For each roadway, a single density value is measured and used as a constant density, with the corresponding natural ventilation pressure and ventilation resistance also being constant values. At this stage, the ventilation system operation can be considered a closed system with an isochoric process. Based on this, the fundamental construction principle of the commonly used airflow regulation models is to treat ventilation resistance and natural ventilation pressure as constant environmental parameters, and to use air volume and adjustment resistance as optimization decision variables. The conservation laws of air volume and air pressure serve as nonlinear equality constraints to complete the nonlinear airflow regulation model setup. In other words, airflow regulation focuses on the dynamic airflow process in the mine, treating the airflow as an isochoric process characterized by constant density.

However, under the influence of unsteady environmental and flow fields, the airflow essentially undergoes variable heat flow under the coupling of aerodynamic and thermal effects, with the state properties of the airflow constantly in a dynamic equilibrium state (referenced from Shengqiang Yang's "Research on thermal and dynamic changes of airflow and thermal resistance in high-temperature and high-humidity mines"). The natural ventilation pressure and ventilation resistance calculated based on the constant state property values of the isochoric process are constant values in time series. Constructing airflow regulation models with constant natural ventilation pressure and ventilation resistance as environmental parameters cannot determine the optimal set of airflow regulation solutions to achieve precise control of ventilation network air volume. Therefore, the core of constructing the airflow regulation model is to establish a equation for the dynamic changes in the airflow state properties (temperature, density, and pressure) in time series.

This paper constructs a transient heat and airflow flow characteristics model and a fluctuating natural ventilation pressure and ventilation resistance model by analyzing the dynamic changes in the state properties of mine airflow in time series, aiming to solve the fluctuating values of natural ventilation pressure and ventilation resistance in time series. By treating fluctuating natural ventilation pressure and ventilation resistance as environmental variables, adjustment resistance as decision variables, and minimizing system power consumption as the objective function, an airflow regulation model based on the heat and airflow coupling characteristics of airflow is constructed.

Compared to traditional airflow regulation model, the airflow regulation model based on the heat and airflow coupling characteristics, which integrates the transient heat and airflow flow characteristics model and the fluctuating natural ventilation pressure and ventilation resistance model, has the following advantages:

1.Consideration of Real-Time Environmental Dynamics: Traditional models typically assume constant state property values for airflow, which is not suitable for the real-time changes in the unsteady environmental and flow fields in actual production mines. In contrast, the model proposed in this paper establishes a transient heat and airflow flow characteristics model based on the dynamic changes in the airflow state properties (such as temperature, density, and pressure) in time series, capturing the changes in airflow state over time and avoiding the limitations of static environmental assumptions in traditional models.

2.Precision Control: Traditional models rely on constant natural ventilation pressure and ventilation resistance for airflow regulation, which cannot adjust according to actual environmental changes, leading to poor accuracy in air volume regulation. The model proposed in this paper dynamically calculates the fluctuating values of natural ventilation pressure and ventilation resistance, using them as environmental variables, combined with adjustable resistance as decision variables, to achieve precise optimization of airflow regulation, allowing real-time adjustment of ventilation network air volume according to actual working conditions.

Based on your suggestion, I have analyzed and summarized the main directions and achievements of existing airflow regulation technology research in Section 1: Introduction, and proposed that "these studies focus on the dynamic airflow process in mines, treating the airflow as an isochoric process characterized by constant density, and constructing and solving airflow regulation models with constant natural ventilation pressure and ventilation resistance as environmental parameters, based on the dynamic properties of airflow." Additionally, I have added a direction for improving existing airflow regulation models in the introduction to lead into the corrected airflow regulation model based on the heat and airflow coupling characteristics. "Natural ventilation pressure and ventilation resistance calculated based on constant state property values of the isochoric process or constant state property values at specific moments are constant values in time series. Constructing airflow regulation models with constant natural ventilation pressure and ventilation resistance as environmental parameters canno

---

## [Editor Report · Decision Letter 1]

18 Feb 2025

Decision on optimal airflow regulation solution set based on heat and airflow coupling characteristics of mine airflow in time series

PONE-D-24-50720R1

Dear Dr. Heng,

We’re pleased to inform you that your manuscript has been judged scientifically suitable for publication and will be formally accepted for publication once it meets all outstanding technical requirements.

Kind regards,

Rajeev Singh

Academic Editor

PLOS ONE
---

## [Editor Report · Acceptance letter]

PONE-D-24-50720R1

PLOS ONE

Dear Dr. Heng,

I'm pleased to inform you that your manuscript has been deemed suitable for publication in PLOS ONE. Congratulations! Your manuscript is now being handed over to our production team.

Kind regards,

on behalf of

Dr. Rajeev Singh

Academic Editor

PLOS ONE